# Do selfies make women look slimmer? The effect of viewing angle on aesthetic and weight judgments of women's bodies

**Ruth Knight** [1] *, **Catherine Preston** [2]

**1** Department of Psychology, York St John University, York, United Kingdom, **2** Department of Psychology, University of York, York, United Kingdom

* r.knight1@yorksj.ac.uk

## Abstract

Taking and posting selfies is a popular activity, with some individuals taking and sharing multiple selfies each day. The influence of the selfie angle, as opposed to more traditional photo angles such as the allocentric images we see in print media, on our aesthetic judgements of images of bodies has not been explored. This study compared the attractiveness and weight judgements that participants made of images of the same bodies taken from different visual angles over a series of four experiments (total N = 272). We considered how these judgements may relate to disordered eating thoughts and behaviours. Selfies were judged to be slimmer than images from other perspectives, and egocentric images were judged to be the least attractive. The way participants rated bodies seen from different perspectives was related to their own disordered eating thoughts and behaviours. These results contribute to our understanding of how we perceive the images we see on social media and how these might be related to how we feel about our own and other people's bodies.

## Introduction

Selfies are synonymous with social media, yet little is known about the effect of viewing selfies on how we feel about our own and others' bodies. Taking and posting selfies is a popular activity, with participants in one study taking over eight selfies a day [1]. Yet research suggests that selfies evoke criticism and are associated with a lack of authenticity and narcissism in the subject [2]. Selfies allow the poster to share social information and have control over photographic and compositional aspects of the picture, thus they might have a strong impact on aesthetic judgements [3]. The connection between selfies and social media is well-established, with many selfies being taken with the intention to post them on social media platforms, such as Instagram, Snapchat, or Facebook [4]. However, selfies make up a relatively small proportion of the photos posted on Instagram (0.7%), so research should also consider how photos taken from other perspectives influence aesthetic judgements [5].

Objectification theory suggests that when someone is objectified, they are perceived as, and consequently behave as, an object instead of an individual person [6]. Social media users may perceive those they see on social media as objects as opposed to people [7]. This could extend

**Data Availability Statement:** All files are available from the OSF here: https://osf.io/n4wqy/?view_only=8f56d534a63c47d697e689e9abb55ee1.

**Funding:** The authors received no specific funding for this work.

**Competing interests:** The authors have declared that no competing interests exist.

to the person posting the content, who begins to see themselves from the observers' view as an object, leading to self-objectification [8]. The objectifying elements found on social media may make us more likely to make upwards comparisons, those that favour others, particularly in terms of slimness and attractiveness, thus damaging our own body satisfaction [9, 10]. Selfies in particular have the potential to be self-objectifying as we are likely to see them from the viewpoint of the people who will be interacting with them, thus perceiving ourselves as a photo to be 'liked' by others. Research indicates that much of the content posted by women on social media is objectifying selfies (selfies containing an objectifying element, i.e., presenting the person in the photo as a sexual object to be viewed by others), and the frequency of posting these images is associated with trait levels of self-objectification [7]. Eating disorders (EDs) have been found to be related to increased self-objectification and objectification of the body [10].

There are several factors that influence attractiveness judgements, particularly in women, including waist to hip ratio (WHR) and overall body fat (indexed by BMI) [11]. There are mixed results in the literature regarding the importance of these factors [11–16]). However, Singh [17] found that WHR was the key variable associated with attractiveness in women, both in terms of personality characteristics and physical attractiveness. Research indicates that an optimal WHR is closer to 0.7 [18].

Selfies have been differentiated from traditional (allocentric) portraits due to the fact that they are taken by the subject of the picture [3]. As such these images have a unique viewing perspective, which does not subscribe to fundamental principles traditionally applied to portrait photography [19] (also see [20]). Schneider and Carbon [3] demonstrated that whether a selfie of the face is taken from above or below and left or right can influence how individuals rate body weight of the subject, alongside other personality characteristics. They found that body weight was judged as higher when the selfie of the face is taken from a lower perspective compared to a higher perspective. The angle of a photograph cannot modulate the actual BMI of the model but may influence the appearance of visual cues that relate to body weight. Although in Schneider and Carbon's study the images were isolating the face and thus would not impact WHR, this is a key weight related cue of the body that could be impacted by visual angle. Selfies can be taken to capture the best angles of the person in the photo, so might make the WHR appear more optimal (closer to 0.7). This may mean that selfies are judged as slimmer and thus more attractive compared to allocentric (traditional media) images.

Lateral selfies gave rise to higher attractiveness ratings than frontal allocentric views of people [3]. However, the study examining this was entirely based on the attractiveness of faces and did not include the contribution of bodies towards these judgements, or whether other visual perspectives used in social media images impact these judgments. Research suggests that on dating applications such as Tinder men are more likely to use selfies taken from below, whereas women are more likely to use selfies taken from above [21]. These results have been partly replicated and extended to include selfies in other contexts such as Instagram. When considering the context that selfies are posted in, women are more likely to post frontal selfies on Tinder compared to Instagram, and men are less likely to post selfies from below on Instagram [22]. This seems to suggest that the context and communication intentions related to selfies influence the angle they are presented from; the authors suggest that selfies are a form of non-verbal communication in this way [22]. On social media, for example, men and women may use selfies taken from below or above respectively due to the role of partner height in attracting potential partners [22]. Related to this is the advent of the selfie-stick. Selfie-sticks are tools that allow us to take selfies from further away. This increased distance may have an impact on aesthetic appraisal of the bodies viewed in selfies, as it enables a greater degree of

variability in angle to create the optimal image. Alongside selfies, allocentric and egocentric views of bodies are also commonly portrayed on social media [23].

The visual perspective from which we view a body may also modulate how bodies are related to the self, for example bodies viewed from a first-person (egocentric) perspective are more easily linked to one's own body [24]. Previous research showed a significant interaction between visual perspective and body size when participants were rating the attractiveness and weight of bodies, in that large bodies were rated as significantly less attractive and larger when seen from an allocentric perspective as opposed to an egocentric perspective [24]. This may represent a self-bias in judgements made towards our own body for larger bodies, or simply due to occlusion of weight-related visual cues from an egocentric perspective. Interestingly there was a non-significant trend of the reverse pattern for slimmer bodies. It has been argued that we tend to make self-promoting judgements of our own characteristics, but we make self-deprecating judgements of our body size. Donaghue and Smith [25] explored whether this extends to other physical attributes, such as attractiveness and sexiness. Participants made self-enhancing judgements of their own attractiveness and sexiness, but self-deprecating judgements of their own body size—they rated themselves as more overweight than ratings made of them by others [25]. If egocentric images are associated with the self, they may be subject to similar bias.

Negative evaluation of our own body alongside positive evaluation of others' bodies as viewed on social media platforms may make us more vulnerable to upwards social comparisons and thus body dissatisfaction. Recent research indicates that viewing more selfies is linked to facial dissatisfaction, a relationship mediated by appearance comparisons such that selfies are associated with facial dissatisfaction through increased appearance comparisons towards these images [26]). However, this effect has not yet been explored in relation to bodies alone.

When we make aesthetic judgements, we typically look at bodies and faces together [27]. However, it has been found that face and body judgements do not interact when individuals make an overall attractiveness judgement; instead, both make significant independent contributions to overall attractiveness [27]. For women, both face and body components influence overall attractiveness, which validates looking at attractiveness judgements of faces and bodies together and individually [27]. Previous research indicates that visual perspective can influence aesthetic judgements of bodies; this work builds on those findings by adding selfie angles and specifically considering social media style images [24]. Some research indicates that faces may provide more information used in attractiveness than bodies [28]. Therefore, as we are interested specifically in how social media style images influence feelings towards the body, we did not include models' faces; this may have resulted in participants rating the attractiveness of the model based more heavily on judgements of their faces as opposed to aesthetic judgements of the body.

There is a substantial body of evidence indicating that there is a link between exposure to images of the thin-ideal, body dissatisfaction and disordered eating [29]. Research indicates that this is the case not just for traditional media, like TV and magazines, but also for social media [30]. It seems that being exposed to appearance related images on social media platforms like Instagram has a detrimental effect on body satisfaction, and is linked to disordered eating, particularly amongst women [31]. The precise mechanisms of this relationship are not clear, but the effect may be related to making more upwards social comparisons when viewing bodies that are deemed more attractive than the viewer's own body [32]. To date there is no research available that has explored the effects of specific kinds of social media image, such as selfies, on body satisfaction and disordered eating. Should selfies be judged in different ways to other images, though, they may have a more powerful influence on body satisfaction and risk for disordered eating.

This study aimed to consider how people judge social media style content of bodies captured from different visual perspectives in a series of four separate experiments. Previous research found that selfies of faces were judged as more attractive [33, 34] and that egocentric images of large bodies were judged as more attractive and slimmer [24]. Based on this research it is hypothesised that selfies will be deemed more attractive and slimmer than both allocentric and egocentric images of the same bodies. We include four different visual angles of the body in an attempt to adequately capture the different kinds of images that are available on social media. Typically, social media platforms include photos taken by another person in an allocentric view, photos taken of one's own body from an individual's own perspective (an egocentric view), and selfies taken either at arm's length or with a selfie-stick. Secondly, based on previous research indicating that participants judge others' bodies as thinner if they have an eating disorder [35], and that social media particularly affects those vulnerable to disordered eating [31, 36] it was hypothesised that aesthetic and weight judgements of social media style body images (particularly selfies) will be related to participants' disordered eating symptomatology (Eating Disorder Examination Questionnaire (EDE-Q) score), such that aesthetically favourable judgements for selfies (attractiveness and slimness) would be stronger for those participants with greater ED vulnerability. Finally, based on previous findings suggesting that WHR and BMI may be important cues in judging attractiveness in women, and given that assumed WHR may change based on the perspective a photo is taken from (unlike BMI), it was hypothesised that differences in WHR between the perspectives will relate to differences in attractiveness judgements across perspectives. Previous research indicates that the optimal WHR is close to 0.7 [37]. Therefore, we anticipate that visual changes in WHR from changes in perspective that are closer to 0.7 will be associated with increased attractiveness judgements. Research indicates that appearance pressures and patterns of body satisfaction differ between men and women, and that body ideals may be more strongly linked to other factors, such as sexual orientation, in men compared to women [38]. Women are typically subject to the thin-ideal, whereas for men muscularity, leanness, or both may be the dominant appearance ideal [39]. Alongside these differences in appearance pressures, there may also be differences in the way men and women present themselves on social media [22]. Due to this, and the relatively larger amount of women who use social media [40, 41] we focus solely on women in this study.

## General methods

### Ethics statement

Ethical approval for all experiments within this study was granted by the Departmental Ethics Board in the Psychology Department at the University of York. Written informed consent was given by all participants who took part.

### Participants

Participants were recruited via adverts on social media and through a departmental system to allow students to participate in experiments to gain course credit. All participants who took part were students at the university. All four experiments had the same inclusion criteria (identifying as a woman, being 18 and over) and exclusion criteria (history of an eating disorder, being under 18).

### Materials

**Measures.** *Eating Disorder Examination Questionnaire 6.0 (EDE-Q).* The EDE-Q is a 28-item self-report questionnaire that assesses eating disorder symptoms in the last 28 days [42, 43]). It traditionally uses four subscales (Restraint, Eating Concern, Shape Concern, and

Weight Concern) as well as a global score, which is calculated from the mean of the four sub-scale scores. However, more recent research indicates that a three-factor model is a better fit for the data, especially in non-clinical samples [24, 39]. Based on this, we use a three-factor model that combines the two Shape Concern and Weight Concern factors into one. Participants rate items on a 7-point Likert scale, with higher scores indicating higher eating disorder psychopathology. There are six items that relate to the frequency of eating disorder attitudes and behaviours in the past 28 days, which do not contribute to the subscale or global scores but provide information on some core eating disorder behaviours such as laxative use and self-induced vomiting. These are not used in this study. This is partly for ethical reasons, as the responses to these questions do not contribute to the subscale scores, and partly as the information collected often provided detailed qualitative responses to the questions. Research has established acceptable levels of internal consistency for global and subscale scores of this three-factor model in men and women, alongside the reliability of the scale [24, 39]. Experiments one to three used this scale.

## Stimuli

Colour photographs were taken of 10 female models' bodies (excluding the head) standing against a white backdrop from different angles on a Samsung tablet. The models were students attending the Psychology department at the University of York and received course credit for taking part. Model BMI ranged from 18.5 to 30.6 (M = 22.46, SD = 4.23). Models were asked to wear form-fitting clothes that they would exercise in. Most models wore leggings and a form-fitting vest top. Models stood with their arms loosely by their sides with their right leg pointed slightly outwards to simulate photos commonly seen on social media. This pose was maintained for every photograph. For the allocentric angle, the experimenter stood roughly two metres in front of the model capturing the whole body. The photo for the egocentric angle was taken by the model by angling the camera down towards the body from just below the chin. The selfie-stick angle photo was again taken by the model by holding the selfie-stick an arm's length away from the body in front of them. The model took the selfie angle with the tablet held in their left hand an arms-length away angled from above (see Fig 1). Experiments one to three used stimuli from the selfie category and one of the three other categories. Experiment four used all stimuli.

## Procedure

Participants accessed all of the experiments via a personalised link through Qualtrics (Qualtrics, Provo, UT) that was sent to their university email address. When following the link, participants first answered demographic questions regarding age, gender, and nationality, followed by instructions for the experimental task. Within each block, they were presented with 10 images of bodies from the same visual perspective in succession and for each image were asked to rate the attractiveness and weight of these bodies on a visual analogue scale (VAS) ranging from 0 to 100 (in separate blocks). The VAS were anchored with 'Very Unattractive' and 'Very Attractive' for the attractiveness ratings, and 'Very Underweight' and 'Very Overweight' for the weight ratings. Participants used their mouse to select the position on the scale that they felt best represented the attractiveness or weight of the body in the photo that was displayed. The image was present on the screen for as long as the participant took to make the judgements. After the judgement was made, the next photo and VAS were displayed. All images were optimized to fit the screen that the participant was viewing them on (participants could access the questionnaire using their smartphone or computer).

Participants judged weight and attractiveness for two perspectives in each of the first three experiments, thus the experiment consisted of four separate experimental blocks: eg selfie

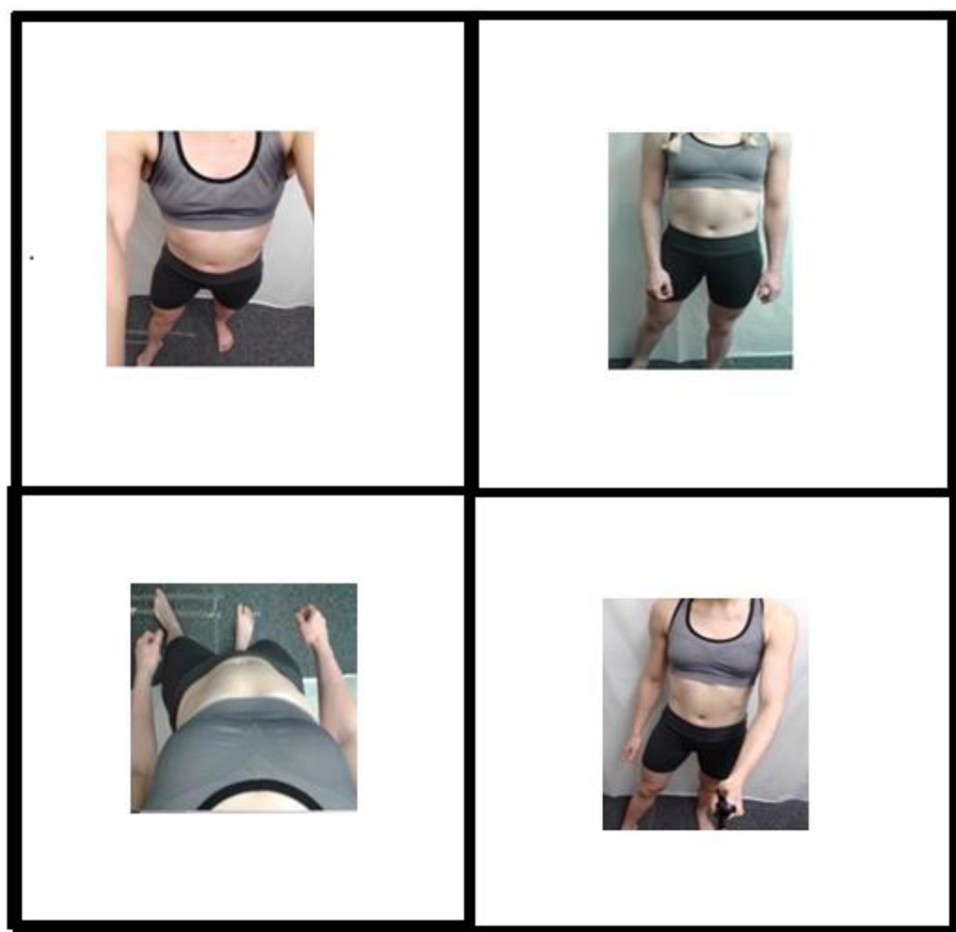

**Fig 1.** Example stimuli: Selfie (top left), allocentric (top right), egocentric (bottom left), and selfie-stick (bottom right) perspectives.

weight judgments, selfie attractiveness judgments, allocentric weight judgements and allocentric attractiveness judgments. The order of these blocks was randomised and images within the blocks were also presented in a random order. Participants saw each image once, thus viewing ten images in each condition/block, 40 trials overall for experiments one to three. In experiment four, participants were shown all images across each condition and were asked to make aesthetic judgements of each one (80 trials overall).

Participants then completed the EDE-Q and recorded their own self-reported weight and height (to calculate BMI) before finishing the questionnaire and debrief. BMI was calculated to compare participants to the general population (we did not compare participant and model BMI.) Weight and height were completed at the end of the study along with the EDE-Q to mitigate any potential effect of answering these body focussed questions on aesthetic judgements during the experiment, aside from for experiment four, for which the EDE-Q was not completed due to time constraints.

## Data analysis

Firstly, in experiments one to three we calculated the EDE-Q subscale scores for each participant by taking the mean of the relevant items for each subscale (Carey et al. 2019 [24]). We

treat the EDE-Q scores as ordinal data, as is the norm in the ED literature (Wu & Leung, 2017 [44]; Jennings & Phillips, 2017 [45]). In line with other studies in the field, we checked EDE-Q scores for normality. We then calculated the separate mean scores for weight and attractiveness ratings for each image within each perspective. This is calculated by taking the mean rating for each stimuli across participant responses, for both weight and attractiveness ratings, so each stimuli had an average weight and attractiveness score across participants. We used paired samples t-tests (or their non-parametric equivalents) to calculate whether the differences in aesthetic ratings were different across perspectives. When calculating normality, we expect that EDE-Q scores and accompanying responses will be non-normal, based on previous studies using a similar demographic. We also implemented non-parametric statistics to reduce the impact of skewed data. For this reason, we do not analyse outliers.

We directly tested the hypothesis that aesthetic and weight judgements will be related to eating disorder symptomatology by assessing if there were significant correlations between EDE-Q scores and the differences in attractiveness and weight judgements between selfies and the other (comparison) perspectives. To this aim, we first calculated difference scores in attractiveness and weight ratings between the two image types.

A power analysis was conducted using R studio (*package pwr*) [46]) based on detecting a medium effect size in paired samples t-tests (as shown in previous studies comparing attractiveness and weight ratings [24]). This suggested that the first three experiments would need at least 33 participants (power = 0.80, alpha of .050, d = 0.50).

We took waist and hip measurements in each photo to calculate WHR. Waist measurements were identified as the width across the body at the smallest visible point of the lower torso. Hip measurements were identified as the widest point of the lower torso below the waist at the top of the thighs. We then tested the hypothesis that WHR will be related to attractiveness judgements, by assessing correlations between differences in WHR and differences in attractiveness judgements across perspectives. Specific details on this process can be found in each experiment's data analysis section.

## Results

### Participants

Table 1 shows BMI and age of the participant samples for each of the four experiments.

**Experiment one–Selfie vs allocentric.** The first experiment specifically aimed to examine differences in attractiveness and weight judgments between selfie images and more traditional allocentric images.

*Experiment one (Selfie vs allocentric) methods.* **Participants.** In total 69 participants meeting the inclusion and exclusion criteria were recruited. All demographic information was taken as self-reported (see Table 1 for sample details).

*Data analysis.* We followed the analysis steps laid out above. To directly test hypotheses that selfies will be judged as more attractive and slimmer than allocentric images, we compared the stimuli across perspectives on attractiveness and weight ratings using pairwise

**Table 1. A table showing means and standards deviations (SD) of age and BMI for each experiment.**

|  | N | BMI (SD) | Age (SD) |
|---|---|---|---|
| Experiment 1 | 69 | 22.59 (3.95) | 19.32 (1.54) |
| Experiment 2 | 50 | 21.72 (3.47) | 18.98 (1.06) |
| Experiment 3 | 44 | 22.61 (3.07) | 19.16 (2.00) |
| Experiment 4 | 109 | 21.82 (3.54) | 19.22 (1.56) |

comparisons. If data were normally distributed, we used paired samples t-tests and Pearson's r correlations. If data were not normally distributed, we used Wilcoxon signed rank tests and Kendall's tau correlations. Bonferroni correction was used for multiple comparisons.

Attractiveness difference scores were calculated by subtracting attractiveness scores for allocentric images from selfie images such that positive scores represented greater attractiveness judgements for selfie images and negative scores represented greater attractiveness for allocentric images. The reverse calculation was done for the weight rating such that positive scores represent slimmer judgements for selfies and negative scores represent slimmer scores for allocentric images.

To examine the relationships between WHR and attractiveness ratings we subtracted WHR for allocentric images from selfie images and correlated these with the attractiveness differences as calculated above. Because the majority of the bodies in our stimuli had a WHR > .07 and we anticipate that selfies would be associated with more optimal WHR, negative correlations between attractiveness and WHR differences would represent relationships in the predicted direction (greater attractiveness would be associated with WHR's closer to optimal in selfie images).

*Experiment one (Selfie vs allocentric) results*. Data for all variables (attractiveness judgements, weight judgements, differences in attractiveness judgements across perspectives, differences in weight judgements across perspectives, and EDE-Q scores) were not normally distributed, according to Shapiro-Wilk tests (largest $p$ value: $p = .021$) and analysis of histograms. Because of this, Wilcoxon signed-rank tests were used.

In line with our hypotheses a Wilcoxon signed-rank test showed a significant difference in weight ratings between allocentric and selfie images (W = 387.00, $p < .001$, rank-biserial correlation = -0.68 [47] such that selfie images were judged to be slimmer than allocentric images (see Table 2 and Fig 2). However, contrary to the hypothesis that selfies would be judged as more attractive compared to allocentric images, a second Wilcoxon signed-rank test showed no significant difference in attractiveness ratings between the conditions (W = 879.50, $p = .073$, rank-biserial correlation = -0.25; see Table 2 and Fig 3).

We calculated means and standard deviations of WHR in the selfie (M = 0.91, SD = 0.05, range = 0.84–1.03) and allocentric (M = 0.80, SD = 0.08, range = 0.68–0.90) conditions. Contrary to the hypothesis that WHR would be related to attractiveness ratings, differences in WHR measured in the images across perspectives (selfie and allocentric) were not correlated with differences in attractiveness judgements ($τ_b = -0.22$, $p = .536$).

The EDE-Q subscales demonstrated good internal consistency in this sample using both Mcdonald's ω and Cronbach's alpha (Shape and Weight Concern ω = .939, α = .937; Preoccupation and Eating Concern ω = .714, α = .708; Restraint ω = 0.764, α = 0.735) [48, 49].

Differences in attractiveness and weight judgements were correlated with the three EDE-Q subscale scores, using Kendall's tau. Contrary to the hypothesis that those with higher EDE-Q scores would judge selfies as slimmer than allocentric images, there were no significant correlations between weight differences and Restriction scores, Shape and Weight Concern scores, or Preoccupation and Eating Concern scores (see Table 3).

**Table 2. Showing means and standard deviations for weight and attractiveness ratings for selfies and allocentric images.**

|                           | M     | SD    |
|---------------------------|-------|-------|
| Selfie Weight             | 45.07 | 7.00  |
| Allocentric Weight        | 48.86 | 5.20  |
| Selfie Attractiveness     | 54.90 | 11.72 |
| Allocentric Attractiveness| 55.58 | 11.74 |

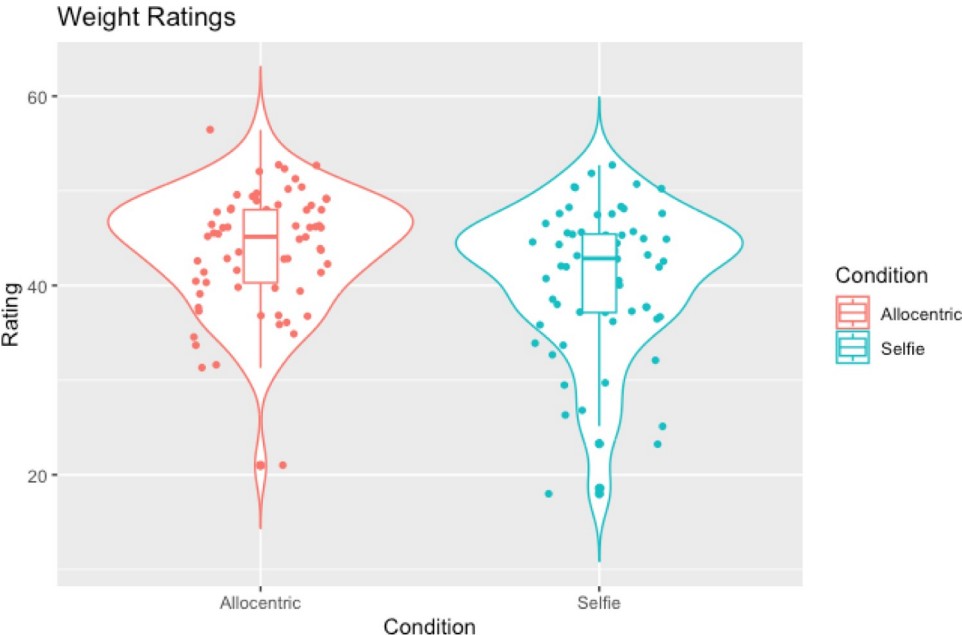

**Fig 2. Violin plots with box plots and data points showing weight ratings across selfies and allocentric images.**

Similarly for the attractiveness ratings, there was no significant correlation found between attractiveness differences and Restriction scores. However, there were significant correlations between attractiveness differences and Shape and Weight Concern scores and between attractiveness differences and Preoccupation and Eating Concern scores (see Table 3), such that greater SWC and PEC scores were related to greater attractiveness ratings for selfies compared

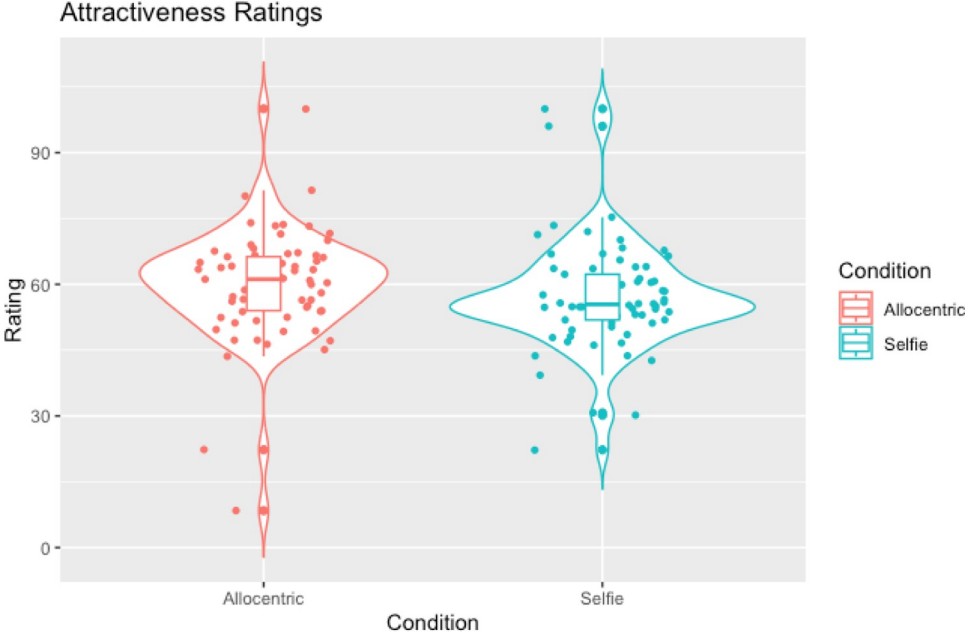

**Fig 3. Violin plots with box plots and data points showing attractiveness ratings across selfies and allocentric images.**

**Table 3. Showing results of correlations between differences in aesthetic judgements across perspectives and EDE-Q subscale scores.**

|  | τb | p |
|---|---|---|
| Weight Ratings Differences with Restriction Score | 0.10 | .262 |
| Weight Ratings Differences with Shape and Weight Concern Scores | 0.14 | .086 |
| Weight Ratings Differences with Preoccupation and Eating Concern Scores | 0.13 | .121 |
| Attractiveness Differences with Restriction Scores | 0.15 | .090 |
| Attractiveness Differences with Shape and Weight Concern Scores | 0.21 | .011 |
| Attractiveness Differences with Preoccupation and Eating Concern Scores | 0.23 | .009 |

to allocentric images. This partially supported the hypothesis that those with higher EDE-Q scores would judge selfies as more attractive.

*Experiment one (Selfie vs allocentric) discussion.* In this experiment we found that selfies were judged to be slimmer than allocentric images, as hypothesised. Contrary to our other hypothesis, we found no significant differences in attractiveness ratings between the selfie and allocentric angles. There was also partial support for the disordered eating hypotheses in that differences in attractiveness judgements were positively correlated with Shape and Weight Concern scores and Preoccupation and Eating Concern scores (although not with Restriction scores). However, there were no significant correlations between weight judgement differences and of the EDE-Q scores. Contrary to our hypothesis, there was not a significant correlation between WHR and differences in attractiveness judgements.

Given the importance of the thin ideal when evaluating attractiveness, it is somewhat surprising that the perceived differences in weight between visual angles did not translate into differences in attractiveness judgements. Interestingly, some research indicates that one's own internalization of the thin ideal and own body satisfaction influences how thinness relates to attractiveness judgements, however this primarily related to mate attractiveness [50]. Research focusing on how women judge the bodies of other women indicates that one's own degree of thin-ideal internalization influences aesthetic judgements of bodies, and thus thinness doesn't necessarily lead to the judgement of being more attractive, at least in healthy participants [51]. Recently we have seen an increased emphasis on leanness, or the combination of thinness and muscularity, when determining attractiveness [52]. It may be that selfies were judged to be slimmer, but muscularity wasn't affected and therefore they were not judged to be more attractive. However, that judgements of slimness were affected could indicate that these images have the potential to be more damaging to those who are already vulnerable to disordered eating. This is supported by the correlations between attractiveness judgements and EDE-Q subscale scores; although across the entire sample selfies were judged as slimmer, it was only for those who have higher eating disorder cognitions and behaviours (weight, shape and eating concern) that this increased perceived slimness is associated with increased attractiveness.

Differences in WHR were not related to differences in attractiveness judgements. This is not surprising given that there were no significant differences in attractiveness judgements across perspectives. Research indicates that an unhealthy weight and WHR is preferred (participants preferred underweight women with an accentuated waist) [53]. Models in this experiment came from across a range of body shapes, and this may have influenced the ways that attractiveness judgements were made about them. Other research suggests that there are multiple attractiveness cues that one considers when making aesthetic judgements, such as thigh girth–height index, waist:chest ratio, height, and BMI, with the latter cues having the greatest influence for some participants [54]. The current experiment only included images from 10 models, which may have been insufficient to detect nuances in the effect of WHR on

attractiveness ratings, given the other potential influencing factors that were we did not measure or were not affected by the change in visual angle.

Research indicates that exposure to social media negatively affects body satisfaction [55]. Given that our results indicate that selfies are perceived as slimmer than allocentric images (the ones most often seen in traditional media), and that there is an established negative effect of viewing traditional media images on body satisfaction [56], it may be that this aesthetic judgement difference could contribute to the detrimental effects of social media on body image. However, images from other perspectives are often seen on social media too, and comparing allocentric images and selfies provides only limited understanding. Egocentric images (pictures taken of the body as if from the location of the eyes in the head) are another example of image types found on social media platforms. Viewing the body from an egocentric perspective is thought to be more readily associated with the self [55]. Previous research indicates that we often make self-biased judgements around attractiveness, such that women tend to judge themselves as less attractive than they believe others would judge them [57]. However, previous studies examining attractiveness ratings of bodies from an egocentric perspective have found images of large bodies to be judged as slimmer and more attractive compared to the same bodies viewed from an allocentric perspective [24]. This might be because physical markers of size for larger bodies are occluded when viewed from an egocentric perspective. In the next study, we therefore compared weight and attractiveness ratings for selfies and egocentric images. Due to the range of body sizes used in the current stimuli (as opposed to large bodies only), we do not anticipate occlusion of weight related visual cues to have a significant impact on weight and attractiveness ratings, instead we predict that selfies will be judged as more attractive compared to egocentric images of the same bodies. This is partly due to negative self-bias [57] and partly due to a positive bias for selfie images.

**Experiment two–Selfie vs egocentric.** *Experiment two (Selfie vs egocentric) methods.* **Participants.** A total of 50 participants were recruited for experiment two. See Table 1 for demographic information.

*Procedure.* The procedure for experiment two was identical to that described for experiment one except that participants were exposed to either selfies or egocentric images.

*Data analysis.* We had an identical analysis plan as described above, this time comparing egocentric to selfie images. Furthermore, to examine the difference in aesthetic and weight ratings for selfies compared to egocentric images related to eating disorder psychopathology and WHR we calculated difference scores in attractiveness and weight ratings between the two image types. Attractiveness differences were calculated by subtracting scores for egocentric images from selfie images such that positive scores represented greater attractiveness judgments for selfie images and negative scores represented greater attractiveness judgements for egocentric images. The reverse calculation was done for the weight ratings such that positive scores represent slimmer judgements for selfies and negative scores represent slimmer scores for egocentric images

*Experiment two (Selfie vs egocentric) results.* According to Shapiro-Wilk tests and examination of histograms, selfie weight averages, selfie attractiveness averages, egocentric attractiveness averages, Restriction subscale scores, Shape and Weight Concern subscale scores, and Preoccupation and Eating Concern subscale scores were not normally distributed (greatest $p$ value: $p = .045$). Because of this, we used Wilcoxon signed-ranks tests and Kendall's tau for analysis.

In line with our hypotheses, Wilcoxon signed-ranks tests showed that selfies were judged as slimmer compared to egocentric images (W = 121.50, $p = .002$, rank-biserial correlation = -0.81; see Table 4 and Fig 4)) and were also judged to be more attractive compared to egocentric images (W = 923.50, $p = .002$, rank-biserial correlation = 0.51; see Table 4 and Fig 5).

**Table 4. Showing means and standard deviations of aesthetic judgements for selfies and egocentric images.**

|  | M | SD |
|---|---|---|
| Selfie Weight Rating | 44.47 | 6.15 |
| Egocentric Weight Rating | 48.70 | 5.02 |
| Selfie Attractiveness Rating | 57.22 | 7.76 |
| Egocentric Attractiveness Rating | 53.38 | 8.78 |

We calculated means and standard deviations of WHR in the selfie (M = 0.91, SD = 0.05, range = 0.84–1.03) and egocentric (M = 0.98, SD = 0.07, range = 0.91–1.08) conditions. As anticipated, differences in WHR were significantly correlated with differences in attractiveness judgements across perspectives (r = -0.64, $p$ = .047), such that more optimal WHR was related to higher attractiveness judgements for selfie images compared to for egocentric images. Given the ranges of WHR for these conditions, this relationship represents increased attractiveness being associated with more optimal WHR measured in the selfie images.

The EDE-Q subscales demonstrated good internal consistency in this sample using both Mcdonald's ω and Cronbach's alpha (Shape Concern and Weight Concern ω = .958, α = .956; Eating Concern ω = .884, α = .875; Restraint ω = .0886, α = .0875).

Difference scores were then correlated with EDE-Q subscale scores, using Kendall's tau. Contrary to the hypotheses that those with high EDE-Q scores would judge selfies as slimmer and more attractive there were no significant correlations between weight differences and EDE-Q subscale scores (see Table 5.) There were also no significant correlations between attractiveness differences and any of the EDE-Q subscale scores (see Table 5).

*Experiment two (Selfie vs egocentric) discussion.* Selfies were judged to be both more attractive and slimmer than bodies photographed from an egocentric perspective, however there were no significant correlations between these differences in aesthetic judgements and disordered eating thoughts and behaviours. This contrasts with experiment one, in which

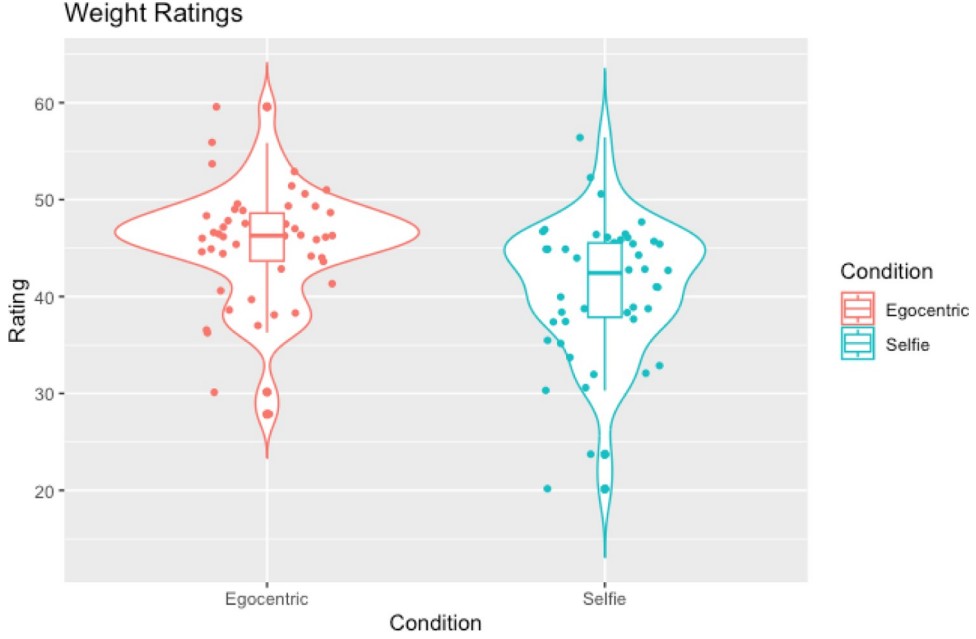

**Fig 4. Violin plots with box plots and data points showing weight ratings across selfies and egocentric images.**

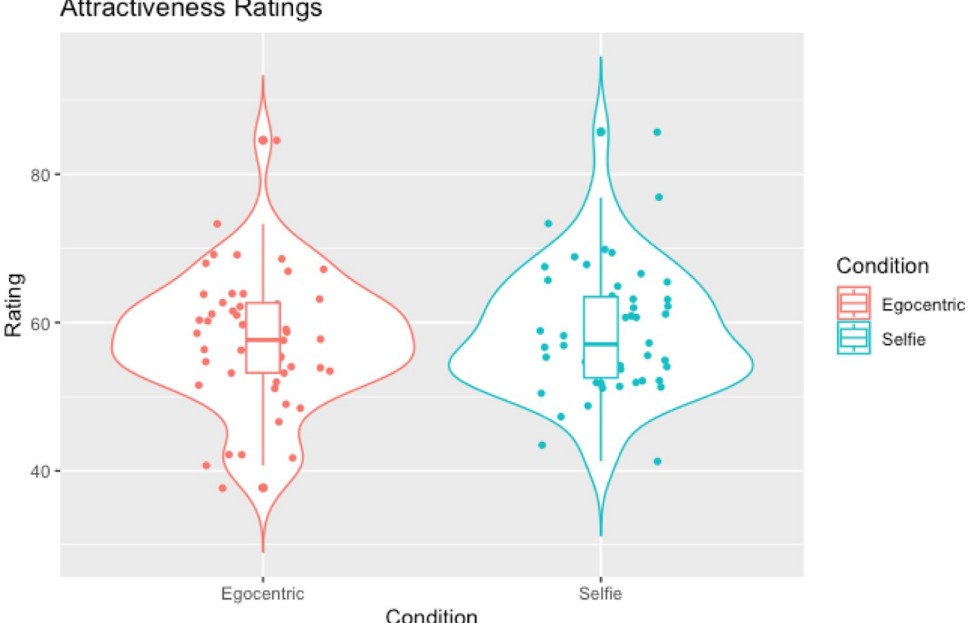

**Fig 5. Violin plots with box plots and data points showing attractiveness ratings across selfies and egocentric images.**

attractiveness differences were related to disordered eating thoughts and behaviours. WHR was significantly correlated with attractiveness judgements, such that more optimal WHR was related to higher attractiveness judgements in selfie compared to egocentric images.

There are various possible explanations for the preference for selfies in terms of attractiveness and weight judgements. Some of these may be similar to the reasons that selfies were preferred to allocentric images in experiment one. Selfies may be a more flattering angle, but they may also endorse typically attractive characteristics such as openness and friendliness [58]. However, because we find a difference in both weight and attractiveness judgments along with an absence of correlation between attractiveness and eating disorder throughs and behaviours as seen in experiment one, this may suggest a slightly different mechanism explaining the results. Thus, we speculate that these results are not primarily driven from a preference for selfies, but from a bias against egocentric images. Egocentric images are taken from the perspective of somebody looking down at their own body, which may therefore be associated more strongly with the self compared to the selfie images of the same bodies. Based on this, a preference for the selfies compared to egocentric images may be in part driven by the negative bias towards the own body that has been found in female participants [57]. If the egocentric

**Table 5. Showing correlations between differences in aesthetic judgements across perspectives and EDE-Q subscale scores.**

|  | τb | *p* |
|---|---|---|
| Weight Differences with Restriction Scores | 0.08 | .458 |
| Weight Differences with Shape and Weight Concern Scores | 0.10 | .307 |
| Weight Differences with Preoccupation and Eating Concern Scores | 0.08 | .430 |
| Attractiveness Differences with Restriction Scores | -0.01 | .899 |
| Attractiveness Differences with Shape and Weight Concern Scores | -0.05 | .645 |
| Attractiveness Differences with Preoccupation and Eating Concern Scores | -0.03 | .747 |

image is associated with the self, participants may have applied this bias and thus judged these images as less slim and attractive. Thus, the higher attractiveness and lower weight ratings for selfies may be due to egocentric images being judged as larger and less attractive. This may also go some way to explaining why selfies were not judged to be more attractive than allocentric images, despite being judged as slimmer. An allocentric image may not tap into the self-bias in the same way that an egocentric image does, thus results might be based on a bias against the egocentric images.

There may also be changes in perceived body morphology driven specifically by the egocentric angle that influences these judgments given that WHR difference between the perspectives related to attractiveness judgments in this experiment. This correlation suggests that the way the body physically appears from an egocentric angle compared to a selfie influences aesthetic judgments made about the body. Putting this into the context of previous results that show a favourable effect of egocentric angles for larger bodies, this might suggest that although weight-related cues for large bodies may be occluded from this viewpoint, other attractiveness cues more important for other body sizes, may appear less optimal viewed egocentrically. Indeed, in the previous study that directly compared egocentric and allocentric perspectives of slim and large bodies, although they found more positive judgments for large bodies from an egocentric perspective, they also found a non-significant trend in the opposite direction (more negative judgments for egocentric images) for slim bodies [24]. The greater variability in body size, along with the different comparison stimuli (selfie) and greater statistical power from the larger sample size in the current experiment, may be why we found significantly lower weight and attractiveness ratings for egocentric images, which was only a trend for slim bodies in the previous study. Similarly, the egocentric angle focuses attention on chest and stomach size, two areas that typically play a role in attractiveness and weight judgements [16, 59]. Having the stomach and chest areas in focus via the egocentric angle may have influenced the way in which participants viewed bodies from this perspective as larger and less attractive. Indeed, such changes in the way the body physically appears from an egocentric angle may contribute to negative self-bias reported in women's judgments towards their own body [57] as well as reports of body size underestimation recorded in overweight participants [60]. Depending on the body size and shape, different aspects of the body may be visible or accentuated, resulting in differing aesthetic judgments at this angle.

Another key difference between egocentric and selfie images is the relative distance from the body from which the image is taken. Research into face perception has shown that the distance the face is from the camera affects attractiveness judgements, with photos of faces taken from further away being deemed as more attractive and associated with more positive personality traits (e.g., trustworthiness) than images of faces for which the photograph was taken close up [31, 34]). Here the egocentric images are taken from much closer to the body compared to selfies. Moreover, when considering the allocentric images from experiment one, it may be that an absence of difference in attractiveness between the two image types could be driven by distance that the photo is taken from the body. Allocentric images are taken at distances further from the body compared to selfies in order to capture the entire body from that angle and due to not being restricted by the length of the arm. This may mean that any positive effect of the selfie angle may be cancelled out by an equivalent positive (aesthetic) effect of the further distance from the body in an allocentric image. It may not be the angle *per se* that is influencing the weight and attractiveness judgments, but the distance that the body is from the camera. However, selfies are not only taken by holding the camera away from you with your arm. As social media and platforms like Youtube developed, along with the tendency to take photos or videos from this perspective, other tools for capturing this kind of content were created. One such tool is the selfie-stick. These are commonly used by content creators whose

images participants may view on social media. The selfie-stick allows for the unique selfie angle with an increased distance from the body. Therefore, the next experiment will compare aesthetic judgements of selfies and images taken with a selfie-stick. This will allow us to compare selfie images of the body taken from close to the body to selfie images taken at from further away. If distance influences perception of the body in the same way as has been recorded for faces, we predict that selfies taken with a selfie-stick would be judged as slimmer and more attractive.

**Experiment 3 –Selfie vs selfie-stick.** *Experiment three (Selfie vs selfie-stick) methods.* **Participants.** For experiment three a total of 44 participants were recruited based on the same power analysis as reported for experiments one and two. For participant demographics please see Table 1.

*Materials.* **Procedure.** The procedure for experiment three was identical to that described for experiments one and two except that participants were shown either selfies or selfie-stick images.

*Data analysis.* An identical analysis plan was used as described for experiments one and two. However, because we predicted that selfie-stick images would be rated more favourably (slimmer and more attractive) compared to traditional selfies, attractiveness difference scores were calculated by subtracting scores for selfie images from selfie-stick images such that positive scores represented greater attractiveness ratings for selfie-stick images and negative scores represented greater attractiveness ratings for selfie images. The reverse calculation was done for the weight rating such that positive scores represent slimmer judgements for selfie-stick images and negative scores represent slimmer scores for selfie images. This means that positive scores were in the direction of our hypotheses, whereas negative scores represent the opposite relationship for both sets of correlations. For the WHR we subtracted WHR for selfie images from the WHR from selfie-stick images. Therefore, negative correlations represented relationships in the direction of our hypothesis (greater attractiveness being associated with more optimal WHR for selfie-stick images).

*Experiment three (Selfie vs selfie-stick) results.* Shapiro-Wilk tests and examination of histograms showed that selfie weight average ratings, selfie attractiveness average ratings, weight rating differences, Restriction scotwres, and Preoccupation and Eating Concern scores were not normally distributed (largest *p* value: $p = .023$). Based on this Wilcoxon signed-rank tests and Kendall's tau were used.

Contrary to the hypothesis that selfie-stick images would result in slimmer ratings compared to regular selfies a Wilcoxon signed-rank test showed that there was no significant difference in weight ratings between selfies and selfie-stick images ($W = 350.50$, $p = .141$, rank-biserial correlation = -0.26; see Table 6 and Fig 6). However, in line with the hypothesis that selfies taken with a selfie-stick would be judged as more attractive than those taken without a selfie-stick, a Wilcoxon signed-rank test showed that selfie-stick images were judged to be more attractive than regular selfie images ($W = 232.00$, $p = .006$, rank-biserial correlation = -0.49; see Table 6 and Fig 7).

**Table 6. Showing aesthetic judgements for selfies and selfie-stick images.**

|  | M | SD |
|---|---|---|
| Selfie Weight Rating | 46.69 | 6.53 |
| Selfie-Stick Weight Rating | 48.17 | 4.40 |
| Selfie Attractiveness Rating | 53.44 | 8.69 |
| Selfie-Stick Attractiveness Rating | 56.52 | 9.62 |

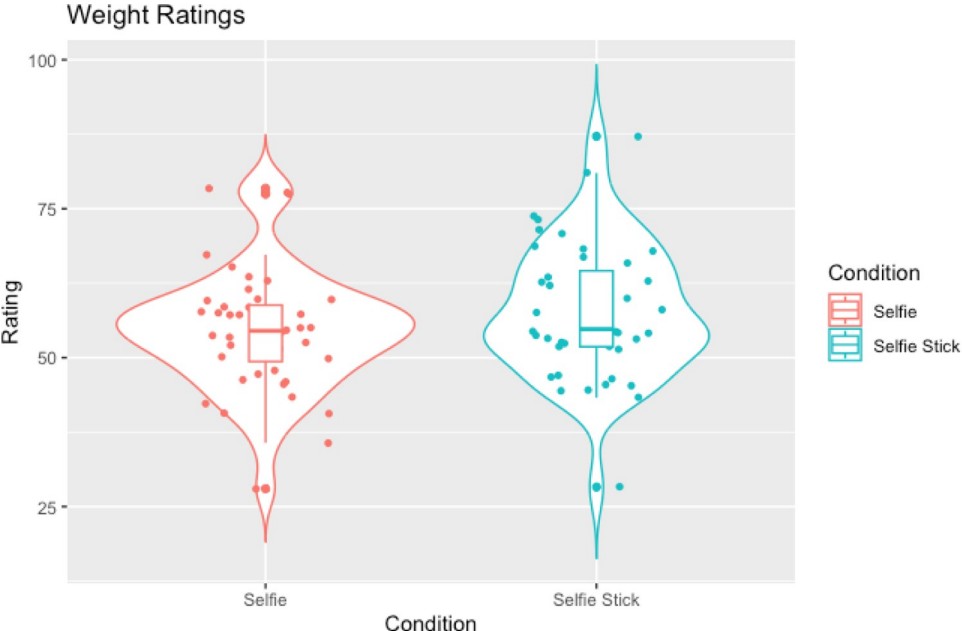

**Fig 6. Violin plots with box plots and data points showing weight ratings across selfies and selfie-stick images.**

We calculated means and standard deviations of WHR in the selfie (M = 0.91, SD = 0.54, range = 0.84–1.03) and selfie-stick (M = 0.83, SD = 0.05, range = 0.74–0.97) conditions. In line with the hypothesis that WHR would be related to attractiveness ratings, differences in WHR measured between the two types of images were significantly correlated with attractiveness

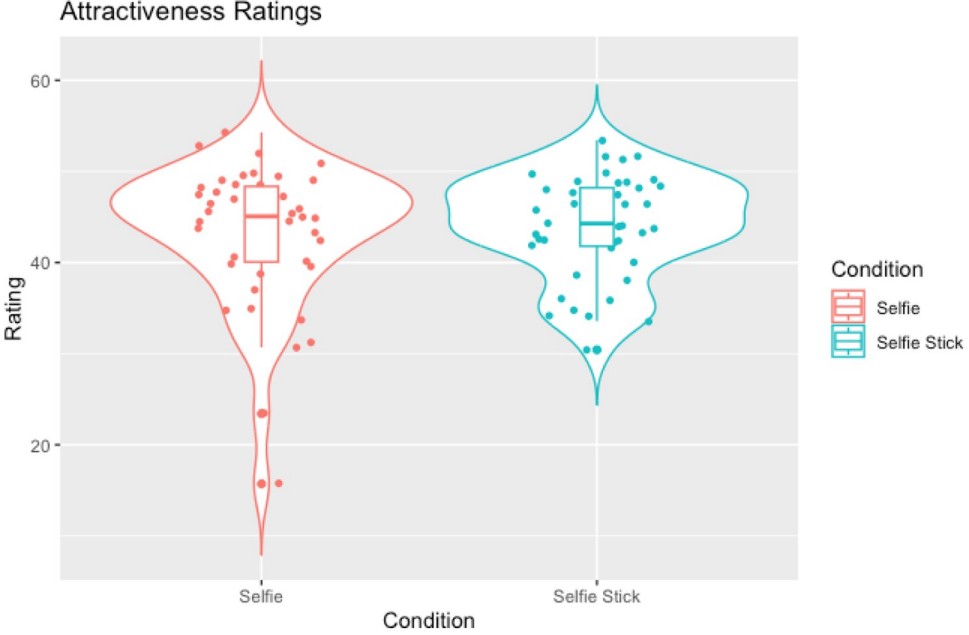

**Fig 7. Violin plots with box plots and data points showing attractiveness ratings across selfies and selfie-stick images.**

**Table 7. Showing correlations between aesthetic judgement differences and EDE-Q subscale scores.**

|  | τb/r | P |
| --- | --- | --- |
| Weight Rating Differences with Restriction Scores | -0.01 | .969 |
| Weight Rating Differences with Shape and Weight Concern Scores | -0.25 | .101 |
| Weight Rating Differences with Preoccupation and Eating Concern Scores | -0.23 | .131 |
| Attractiveness Rating Differences with Restriction Scores | 0.35 | .022 |
| Attractiveness Rating Differences with Shape and Weight Concern Scores | 0.16 | .311 |
| Attractiveness Rating Differences with Preoccupation and Eating Concern Scores | 0.24 | .119 |

scores (r = -0.71, *p* = .022), such that more optimal WHR was related to higher attractiveness judgements for selfie-stick images compared to for selfies.

The EDE-Q subscales demonstrated good internal consistency in this sample using both Mcdonald's ω and Cronbach's alpha (Shape Concern and Weight Concern ω .926, α = .922; Eating Concern ω = .834, α = .803; Restraint ω = .0825, α = .0811).

Weight and attractiveness difference scores were correlated with subscale scores of the EDE-Q. There were no significant correlations between weight differences and Restriction scores, between weight differences and Shape and Weight Concern scores, or between weight differences and Preoccupation and Eating Concern scores (see Table 7). There were no significant correlations between attractiveness differences and Shape and Weight Concern scores or attractiveness differences and Preoccupation and Eating Concern scores, however, there was a significant correlation between attractiveness differences and Restriction scores such that higher Restriction scores were associated with greater attractiveness ratings for selfie-stick compared to regular selfie images (see Table 7).

*Experiment three (Selfie vs selfie-stick) discussion*. In this experiment, selfie-stick images were judged to be more attractive than selfies, but there were no differences in weight judgements across perspectives. There were significant correlations between differences in attractiveness scores across perspectives and the amount of restriction-related thoughts and behaviours participants reported. There were no significant correlations between differences in weight judgements and disordered eating, which is unsurprising given that there were no differences in these judgements across perspectives. WHR was correlated to attractiveness judgements.

Research into the difference between traditional selfies and those taken via a selfie-stick is extremely limited. Some research indicates that those taking selfies are judged to be more socially attractive than those taking selfies with selfie-sticks [61]. However, we are not able to find any other literature exploring the aesthetic judgements of selfies taken in different ways. It is, therefore, difficult to understand why selfie-stick images are deemed more attractive than traditional selfies. The results are in line with the idea that increased distance from the body increases attractiveness in a similar way to that described for faces, but this is not due to changes in overall body size, given the lack of significant difference between the perspectives for weight judgments. Alternatively, the increased attractiveness rating for selfie-stick images may be related to the difference in the visual angle between selfies and selfie-stick images. The added height/distance afforded by a selfie-stick may influence physical cues visible on the body, allowing for a more optimal appearance. Although this does not seem to be associated with any difference on overall body size, this is supported by significant correlations between differences in attractiveness and WHR for selfie and selfie-stick images. This suggests that selfie-sticks allow for a more optimal WHR which leads to increased judgments in attractiveness for these images.

Interestingly, there was also a significant correlation between restriction and increased attractiveness judgments for self-stick images, such that those with high rates of restriction gave higher attractiveness judgments for selfie-stick compared to selfies. Exactly why individuals with high scores on this particular subscale would judge selfie-stick images as more attractive is unclear. The restriction subscale captures engagement in behaviours to change body shape. Although this is traditionally linked to slimness, recent work has also linked this to muscularity [40]. Therefore, it might be that individuals with this type of mindset focus on other body cues (such as WHR) as opposed to overall thinness in order rate attractiveness. However, this is highly speculative and future research is needed to further understand these results.

As in the previous experiments, differences in attractiveness judgements may highlight important differences in the ways that social media content influences viewers. If social media users are more likely to judge certain kinds of content, specifically selfies, as attractive then it may be that this particular type of content is more likely to elicit upwards comparisons and thus body dissatisfaction. In this collection of three experiments selfies have been compared to images from one other angle. On social media, users are likely to see images of bodies from multiple angles in one browsing session. It seems sensible, then, that the final experiment in this series compares aesthetic judgements across all four perspectives in one session, to replicate the above described findings.

**Experiment four–Selfie vs. selfie stick, allocentric and egocentric images.** *Experiment four methods.* **Participants.** In total 109 participants took part in the experiment using the same inclusion/exclusion criteria as described for experiments one, two and three. Age and BMI information can be seen in Table 1.

*Materials.* **Stimuli.** Stimuli for this experiment were identical to that used in experiments one, two and three, such that in this experiment stimuli across each of the four conditions (allocentric, egocentric, selfie, and selfie-stick) were used. See Fig 1 for examples of the stimuli.

*Procedure.* The procedure for the experiment was the same as the initial three experiments, however due to time constraints, the EDE-Q was not included in this experiment. In this experiment participants saw all images from all perspectives, totalling 80 trials in all.

*Data analysis.* In this experiment we aimed to replicate our earlier results that supported the main hypotheses that selfies will be judged as slimmer than both allocentric and egocentric images. We hypothesised that selfies will also be judged as more attractive compared to egocentric images and that selfies taken with a selfie-stick will be judged as more attractive compared to regular selfies. To analyse this we used two one-way repeated measure ANOVAs. One ANOVA was for weight judgements and a second for attractiveness judgements. Power analysis (Rstudio, *pwr package;* [46]) based on a medium effect size for a general linear model analysis including four conditions, suggested that at least 77 participants were needed for the experiment (power = 0.80, alpha of .050, f = .15).

As this experiment was designed to replicate our previous findings from experiments one to three, we decided to include Bayesian analysis for all non-significant results to determine the likelihood of the data occurring under the null hypothesis [62]. The BF represents a likelihood ratio of the alternative relative to the null hypothesis [63]. A BF > 3 indicates evidence for the alternative hypothesis, a BF < 0.33 indicates evidence for the null hypothesis and a BF between 0.33 and 3 being inconclusive [63].

*Results–Experiment four weight judgements.* To examine difference in weight judgments across perspective we conducted a repeated measures one-way ANOVA, with a single factor of perspective (allocentric, egocentric, selfie, selfie-stick). Data met the assumption of being normally distributed according to examination of histograms and Shapiro-Wilk test (all *p* values > .050). Mauchly's test indicated that the data violated the assumption of sphericity ($p = .043$) thus the Greenhouse-Geisser correction was used.

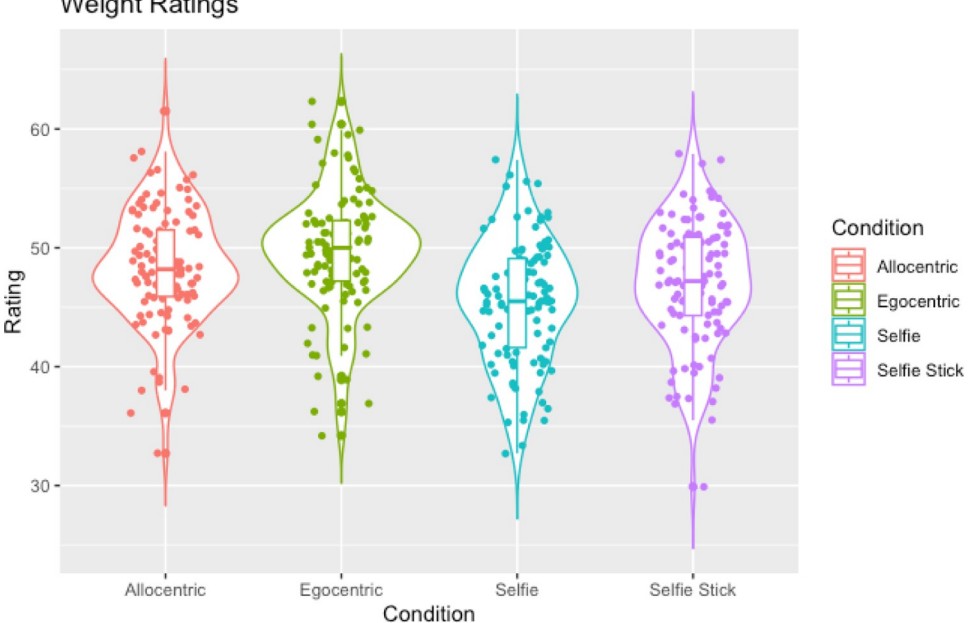

**Fig 8. A violin plot showing weight ratings across the perspectives with box plots and individual data points.**

The ANOVA revealed a significant effect of perspective on weight ratings (F(2.79,300.88) = 27.29, $p < .001$, $\eta p^2 = 0.20$). We used pairwise comparisons to follow this up with Bonferroni correction (critical $p = .0167$). Firstly, we examined whether selfies were judged as slimmer compared to allocentric images as was found in experiment one. Supporting this hypothesis, we found that selfies (M = 45.50, SD = 5.20) were judged as slimmer compared to allocentric (M = 48.45, SD = 4.8) images (t(108) = -6.04, $p < .001$, d = .58). Next, supporting the findings from experiment two, selfies (M = 45.50, SD = 5.20) were judged as significantly slimmer compared to egocentric (M = 49.79, SD = 5.10) images (t(108) = -7.31, $p < .001$, d = .7). Finally, contradictory to the results of experiment three, selfies (M = 45.50, SD = 5.20) were judged as slimmer compared to images taken with a selfie-stick (M = 49.79, SD = 5.10) (t(108) = 2.98, $p$ = .004, d = .29). More information can be found in Fig 8, which shows that selfies were rated as the slimmest, followed by selfie-stick images, with egocentric and allocentric images rated as the largest.

*Attractiveness judgements.* To examine the effect of perspective on attractiveness ratings we conducted a one-way repeated measures ANOVA on attractiveness VAS scores, with the single factor perspective (allocentric, egocentric, selfie, selfie-stick). Data met the assumption for, normality (Shapiro-Wilk $p$-values all > .050) and did not violate the assumption of sphericity ($p$-values > .050).

The ANOVA for attractiveness judgments revealed a significant effect of perspective on ratings of attractiveness (F(3, 324) = 14.67, $p < .001$, $\eta p^2 = 0.12$). In order to replicate findings from experiments one, two and three we conducted pairwise comparisons addressing our main hypotheses. Bonferroni correction was used with a critical $p = .0167$. Firstly, we wanted to examine whether selfies (M = 53.31, SD = 8.23) were judged as more attractive compared to allocentric images (M = 54.91, SD = 8.55). Replicating the findings from experiment one, this comparison was found to be non-significant, with an inconclusive BF (t(108) = -2.25, $p$ = .026, d = .22, BF = 1.19). Next, as anticipated from the results in experiment two, selfies (M = 53.31, SD = 8.20) were judged as more attractive compared to egocentric (M = 50.65, SD = 7.70) images (t(108) = 3.65, $p < .001$, d = .35). Contrary to the findings of experiment three,

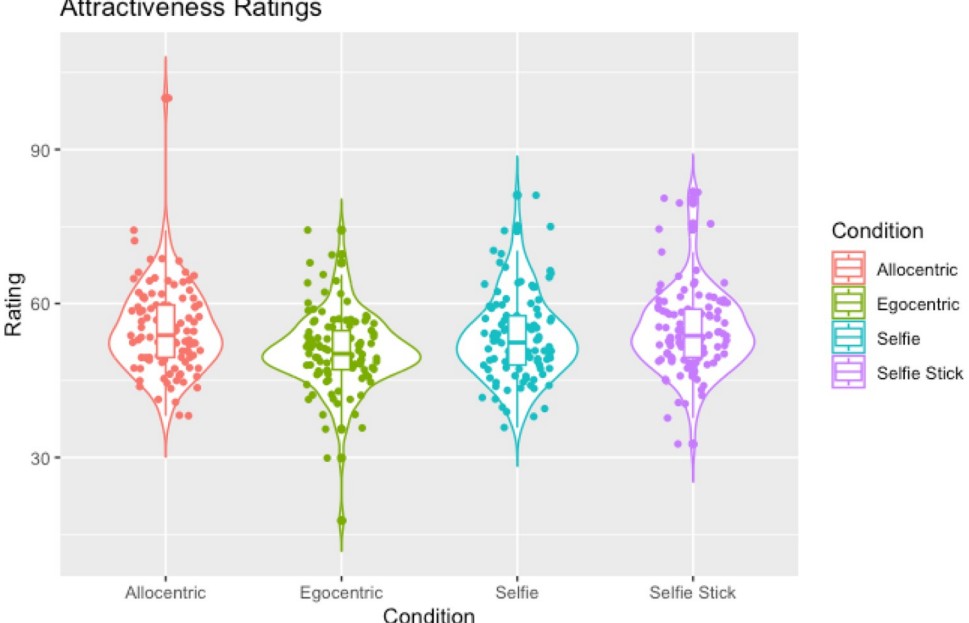

**Fig 9. A violin plot showing attractiveness ratings across each perspective with box plots and data points.**

however, the difference between attractiveness ratings for selfies (M = 53.31, 8.23) and selfie-sticks (M = 54.70, SD = 8.26) did not reach significance when correcting for multiple comparisons and this comparison also had an inconclusive BF (t(108) = 2.18, $p$ = .031, $d$ = .35, BF = 1.03). More information can be found in Fig 9, showing that egocentric images were rated as the least attractive compared to the other perspectives.

*Experiment four discussion.* Selfies were judged to be the slimmest, followed by selfie-stick images, and egocentric and allocentric images were judged to be the least slim. Egocentric images were judged to be the least attractive, compared to all other three perspectives (allocentric, selfie, and selfie-stick). These results replicate the results of experiments one (selfie vs allocentric) and two (selfie vs. egocentric), but diverge from findings reported for experiment three (selfie vs. selfie-stick). Results indicate that egocentric images are judged to be less attractive than selfies and that there was no significant difference in attractiveness ratings between selfies and allocentric images, but do not support differences in attractiveness ratings between selfies and selfie-stick images. These results also replicate findings that selfies are judged as slimmer than egocentric and allocentric images, however interestingly in this experiment selfies were also judged to be slimmer than selfie-stick images, contrary to what was found in experiment three. As EDE-Q responses were not collected in this experiment due to time constraints, it is not possible to ascertain the links between these difference in aesthetic judgements and disordered eating thoughts and behaviours. BF were inconclusive for the nonsignificant comparisons. Although this does not provide statistical support for the null hypothesis, these results also do not support the alternative hypotheses, such that any effect of perspective in these comparisons is likely to be very small.

Taken together with the results of the preceding experiments, these results suggest that visual angle a photograph is taken from does have an impact on the aesthetic judgements we make of bodies, and that it is not slimness alone that underlies attractiveness judgements. Instead, attractiveness may be related to other factors and the social characteristics we ascribe to those taking selfies [61] as well as visual changes in other bodily cues relating to weight and

shape, such as WHR. Interestingly, these results also demonstrate that differences in weight and attractiveness judgments between selfies taken with and without selfie-sticks are less robust. This makes sense given that the angles of these two types of images are very similar compared to differences in appearance of the body between selfies and the other comparison perspectives. Moreover, even if some aspects of the of the body in selfie-stick image may appear marginally more optimal, social characteristics associated specifically with this tool may counteract this, at least for some individuals. Those who use selfie-sticks are judged as less socially attractive than traditional selfie takers [40]. This more negative association with selfie-sticks may influence size ratings as research indicates that social attractiveness is associated with slimness [64]. This may suggest a more complex combination of factors influencing our judgments of selfie-stick compared to selfie images that could be influenced by participant individual differences and thus explain different patterns of results using different population samples. However, considering the dearth of research specifically on self-sticks, such explanations a largely speculative.

## General discussion

This study aimed to consider whether the perspective that images of the body are taken from influences how we judge their attractiveness and weight, including social media style images such as selfies. We also wished to consider whether any such differences are influenced by disordered eating thoughts and behaviours. It was hypothesised that selfies would be judged as more attractive and slimmer than other images, and that selfies taken with a selfie-stick would be judged as slimmer and more attractive than regular selfies. It was also hypothesised that the role perspective plays in aesthetic judgements would be related to disordered eating thoughts and behaviours. Finally, it was anticipated that apparent WHR differences between the different angled images would relate to attractiveness judgements across perspectives.

### Differences in attractiveness and weight judgements across perspectives

Results suggest that selfies are viewed as being slimmer than both allocentric images, which are associated more with traditional media, and egocentric images, which are thought to be more linked to the self [24]. Selfies are widespread on social media, and there may be a negative impact on body satisfaction of viewing these images [55]. Our results suggest that this may be due to bodies in selfies appearing slimmer and thus leading to more upwards social comparisons. The effect sizes for these differences in weight judgements were medium to large across our experiments, suggesting that the effect of selfies being viewed as slimmer is not trivial. However, in contrast, effect sizes for attractiveness ratings were notably smaller and thus may reflect a more complex picture for aesthetic judgments across image angles as opposed to simply that bodies that appear slimmer are also being judged as more attractive.

In terms of attractiveness differences, selfies were judged as being more attractive than egocentric images. When comparing all the perspectives in experiment four, egocentric images appeared to have the lowest attractiveness judgments compared to images from the other three perspectives. In contrast to these results, a previous study showed that large bodies were judged as more attractive from an egocentric perspective [24]. However, this was only with large bodies whereas the current study contains a range of body sizes. This potential interaction between visual perspective and body size in relation to varying perspectives may explain the lack of the hypothesised difference in attractiveness ratings between selfie and allocentric images; the pattern might be different depending on the size of the body [24], therefore future studies should specifically examine how difference perspectives influence appearance and appraisal of different body types.

It may also be that participants viewed selfie images as being more attractive due to the qualities that they convey or are associated with. Selfies have been suggested to endorse the idea that the person in the photo is more extroverted, sociable, and open to experience; qualities that have previously been associated with attractiveness [65, 66]. However, selfies have also been linked to more negative personality traits; such as narcissism and untrustworthiness [67, 68] which may partly dissociate a link between slimness and attractiveness. Interestingly, significant difference in slimness between perspectives did not always have a corresponding significant difference in attractiveness ratings in these experiments. This may suggest that the link between slimness and attractiveness is not as clear as anticipated, and other factors, such as assumed personality traits and related judgements, may play some role in these aesthetic values. It may be, for example, that weight judgements are predicated more on bottom-up perceptual processes, and attractiveness judgements are linked to top-down components too. Further research can consider these questions in greater depth.

In experiment three, selfie-stick images were judged to be more attractive than typical selfie images. This may be related to the photo being taken from further away. Research into face perception has shown that the distance the face is from the camera affects attractiveness judgements of the face, with faces seen from further away being deemed more attractive than those seen from closer [33, 34]. Therefore, it may be that distance from the camera could influence attractiveness judgements of bodies in a similar way. Selfie-sticks placing the subject's body further away from the camera compared to traditional images; this may explain judgments of greater attractiveness for selfie-stick images. The mechanism by which increased attractiveness of the face is thought to be associated with increased distance is that when further away faces appear more convex and so may appear as rounder [33]. Whereas this may soften features of the face, if the same effect is also applied to bodies, it could make bodies appear larger and less socially desirable (slim with flat stomach). Another potential mechanism though which a selfie-stick may influence attractiveness ratings is through the effect of perceived social characteristics. The enhanced selfie-angle achievable with a selfie-stick may also enhance the social cues linked with attractiveness (appearing smaller) [21] thus increasing the individual's rated attractiveness. These competing observations may explain why experiment three only found increased attractiveness and not slimness for selfie-sticks compared to regular selfies; the body may appear rounder and less slim, but the social attributes linked to the appearance and position may make the image more attractive. Those seen using selfie-sticks have also been judged as less attractive than individuals taking selfies without selfie-sticks [61] and increased attractiveness for selfie-sticks was not replicated in experiment four, instead regular selfies were judged to be slimmer with no difference in attractiveness ratings. There may be many potential competing factors influencing how images taken with selfie-sticks are judged. This is also a type of image that has received very little research focus. Perhaps differences between selfies and selfie-stick images are not robust and may depend on the individual characteristics of the person depicted in the image as well as the person rating the image. Further research comparing selfies and selfie-stick images is needed to elucidate the differences between these kinds of images, alongside studies examining distance effects on judgements and appearance features of the body (not just the face).

The differences that were observed in weight ratings between selfies and the other perspectives may be driven by the difference in visible appearance of the body from this angle. This may be linked to the aforementioned effects in face perception experiments, namely that faces appear convex when seen from further away and flatter when seen from closer [33]. Selfies (without a self-stick) are closer compared to allocentric images and thus may be seen as less round, which may make them appear slimmer. The findings that selfies are judged to be slimmer may also be related to selfies traditionally being taken from above the subject (from the

front), which could make the body shape more optimal in terms of conforming to social ideas (making the body appear slimmer). Similarly, selfie angles may provide an optimal WHR compared to traditional allocentric media images, which is suggested as a possible cue for attractiveness and overall body fat [69], however, we did not find any correlations between differences in weight ratings and differences in WHR for any of the examined comparisons. Negative correlations with WHR were only found for attractiveness ratings comparing selfies with allocentric and selfie stick images. The correlations with apparent differences in WHR (being more optimal) and (higher) attractiveness ratings supports the idea that the different visual angles directly affect visual body cues relevant to body size and attractiveness. However, sample sizes for these analyses were small and should be interpreted with caution. We did not analyse correlations between slimness and attractiveness judgements, as this was beyond the scope of the current study, which aimed to consider differences in aesthetic judgements across perspectives. However, in the future researchers may want to explore whether or not the differences in attractiveness judgements are driven by perceived slimness, or another contributing factor.

## Aesthetic judgements and disordered eating

Our results suggest that aesthetic judgments of selfies may be related to some of the links between social media use, body dissatisfaction, and disordered eating, such that having higher levels of disordered eating thoughts and behaviours is related to more favourable (at least in terms of body appearance in relation to the social ideal) judgements of social media style selfies. Positive correlations between differences in aesthetic judgements that favour selfies compared to allocentric and egocentric images and disordered eating symptoms were predominantly related to thoughts and behaviours to do with weight and shape concern. Research suggests that we often compare our own appearance with the photos we see on social media [70, 71]. If selfies are deemed as slimmer and therefore more desirable due to being in accordance with the thin ideal, then we are more likely to make damaging upwards comparisons to those images on social media [72]. This could lead to increased body dissatisfaction, which is a risk factor in developing disordered eating [73]. This may mean that, as selfies are judged as being slimmer, they could be more damaging to those vulnerable to developing an eating disorder, who might give more prominence to weight in judgments of attractiveness.

An increased damaging effect of selfies to those who are already vulnerable is supported by results from experiment one. In this experiment selfies were judged as significantly slimmer compared to allocentric images, but with no overall significant effect of attractiveness. Instead, significant positive correlations were found between shape and weight and eating concerns with increased attractiveness ratings for selfies. This may indicate that slimmer judgments for selfies only influenced attractiveness ratings for individuals more vulnerable to body dissatisfaction. Those with high body dissatisfaction are thought to put more emphasis on appearance as opposed to personality. This may mean that physically appearing slimmer in a selfie out ways any influence of this angle on personality characteristics for these individuals. This is important because it may help explain why those vulnerable to body satisfaction have been found to have more negative effects of social media [36] and also implicate certain types of images (i.e. selfies) may be more damaging than others. This may also mean that even though our significant findings with attractiveness had only a small effect size (as opposed to medium and large effects for finds regarding weight judgments), these effects may disproportionally effect those who already have low body satisfaction. We collected information on participant weight and height to calculate BMI, which is standard practice when asking participants to respond to EDE-Q questions. Some participants had BMI that would be categorised as

underweight. We did not incorporate this information into analysis, but future research may wish to consider the effect of participant BMI when making aesthetic judgements of social media images, particularly relating to our hypothesis that those who are already more vulnerable to disordered eating thoughts and behaviours might be more negatively affected by this kind of content.

There are several limitations to this series of experiments. Initially two perspectives were compared with each other in three separate experiments, as well as collecting EDE-Q scores. In the final experiment however, participants saw images from all four perspectives, but did not fill in the EDE-Q. It would have potentially been more effective to recruit a larger number of participants who saw all stimuli and also completed the EDE-Q so we could attempt to replicate not only the effects of perspective, but also the positive correlational relationships with eating disorder thoughts and behaviours. It is important to note, too, that although for weight-related judgements effect sizes were medium-sized, effect sizes for attractiveness judgments were only small, with small correlations found as well. The small effect sizes remind us that, although participants may judge the attractiveness of social media style images differently based on the perspective that they are taken from, these differences may not have much impact in terms of vulnerability to developing disordered eating or poor body satisfaction for the majority of social media users. However, even small effects often disproportionately affect those most vulnerable [36], which is also supported with our own findings, and thus may still be important to consider especially in relation to mental health issues such as EDs [74, 75].

When conducting experimental research, it is important to consider ecological validity, particularly as previous research in the area has been criticised for not being sufficiently ecologically valid and presenting stimuli in isolation from their usual context [76]. Making aesthetic judgements in an experiment may not be the same as making aesthetic judgements in everyday life. Research has explored whether the attractiveness judgements for bodies individually differ to those given when seen amongst other bodies, and no significant difference was found, suggesting that bodies have an attractiveness value regardless of whether or not they are viewed and judged alongside other bodies [77]. This suggests that the attractiveness judgments participants make in experimental conditions are not necessarily different from those we make in everyday life, thus allowing us to assess these questions using simple experimental designs isolating images of bodies. However, in practice social media users more rarely see bodies depicted without faces (although this does happen, particularly through fitspiration accounts and content). This may have affected the ecological validity of this series of experiments, as we asked participants to make judgements of bodies without faces. Future research should consider whether similar effects as those found in this study are also found when stimuli are images of bodies with faces included in the image.

As mentioned in the data analysis section, we recruited more participants than needed in anticipation of attrition. However, although we recruited enough participants to be somewhat over-powered to detect effects in the pairwise comparisons, we were not sufficiently powered in terms of detecting correlations, particularly for correlations with WHR. For WHR, due to limitations concerning the number of stimuli, we were only powered enough to detect large directional relationships ($r > 0.71$), and for correlations with EDE-Q scores we could detect correlations at $r > 0.29$ for experiment one, $r > .34$ for experiment two, and $r > .37$ for experiment three (all calculations were made using the RStudio pwr package [46]). This may go some way to explaining why we did not detect the relationships we anticipated around WHR and attractiveness judgements, and why we did not detect relationships between aesthetic judgements and some of the EDE-Q subscale scores. However, differences that were observed between the pattern of results do at least reflect a difference in magnitude of effect between conditions. In addition, for many of our null correlations the coefficients were close to zero,

particularly concerning potential relationships between the EDE-Q and differences between selfie vs. egocentric images. Future investigations into these relationships should ensure that sufficient participants are recruited to detect even small effects, given the complex nature of the phenomena in question.

Another limitation relates to the lack of standardisation in the stimuli. Photos were taken a set distance away from the model (for allocentric and egocentric images), an arm's length away from the model (selfie) or the length of a selfie-stick away from the model. This was done to try to create ecologically valid stimuli, by producing photos that were similar to those that participants may typically see on social media platforms like Instagram. To control these parameters as much as possible, we set a standard distance from which the allocentric image was taken, set a specific level of the shoulders from which the egocentric picture was taken, set the selfie as taken from the model's arm length away, and set a standard length of selfie-stick from which the selfie-stick images were taken. We focused on developing stimuli that were as close as possible to typical Instagram images, however, the precise angle or distance from which the photos were taken may have influenced results. Furthermore, the images were optimized for the screen they were viewed upon; the way these would have been seen on a smartphone would have differed to how they would have been viewed if participants used a desktop computer or a laptop. This may have influenced results as not only would the images be a different size, but they may also have been viewed at different brightness and other relevant screen settings. There is no available research regarding whether these factors influence our aesthetic judgements of images, so it is difficult to ascertain how this may have affected results. The results of this study indicate that the perspective from which a photo is taken does influence attractiveness and weight judgements, so future research should explore whether this is modulated by the distance and location the photo is taken from, or whether it is specific to different perspectives. That only female participants took part in this study could also limit our findings somewhat. We focused on women's aesthetic judgements as all of the models for the stimuli were women and we were interested in potential impact on social comparisons and body satisfaction. It may be that ratings of these stimuli from non-female participants would differ. Future research should consider whether there is an impact of model and/or participant gender on aesthetic judgements. Furthermore, we did not consider whether participant BMI was associated with aesthetic judgements, particularly in terms of egocentric images and the preferences for selfies. Meta-analysis indicates that participants in larger bodies are less accurate in body size estimations, which may be due to body dissatisfaction or different somatosensation [78]. Based on this, future research might consider how body size of the observer influences aesthetic judgements.

This research may have implications relevant to clinical practice. The relationship between social media and body satisfaction has received much research attention, but there has been less focus on the specific characteristics of social media content that may be having an effect on feelings towards the body. These results highlight that particular kinds of content may influence body satisfaction in different ways based how viewers judge the attractiveness and weight of the bodies represented in it. Those who have a pre-existing vulnerability to disordered eating may be more likely to be negatively influenced by images of bodies that they deem to be more attractive or slimmer than their own and thus that selfies may be particularly detrimental. Clinicians and educators may be able to use these results to help educate social media users around the kinds of effects different social media content may have on their body image. Specifically, our findings suggest that any preventative or educational measures applied to social media not only should consider type of content in terms of broader categories (e.g., body vs. non-body content) but also the nature of visual angle that bodies that are represented. For example, increasing awareness that the bodies we see on social media may appear slimmer

than they are in reality, particularly in selfies, may help to lessen any negative impact of these images. Awareness campaigns around the high numbers of filters applied to many images found on social media have helped to give content consumers knowledge that the individuals portrayed in these images most likely do not look that way in reality. Thus, awareness of the potentially misleading effects of visual angle (selfie) may do the same, helping users to interpret the images more accurately.

In conclusion, this series of studies aimed to explore the influence of the perspective that social media style images are taken from on attractiveness and weight judgments of bodies. Selfies are judged to be slimmer than other perspectives, and photos taken from the egocentric perspective are judged to be the least attractive. It also seemed that increased disordered eating thoughts and behaviours were related to judging selfies to be more attractive. This calls into question whether viewing images could have a significant detrimental effect on how the viewers feel about their own body and how important these images are for eating disorder vulnerability. Furthermore, future research should aim to consider these questions using ecologically valid paradigms relevant social media platforms used in everyday life to ensure these effects are not just specific to experimental environments.

## Author Contributions

**Conceptualization:** Ruth Knight, Catherine Preston.

**Data curation:** Ruth Knight, Catherine Preston.

**Formal analysis:** Ruth Knight, Catherine Preston.

**Investigation:** Ruth Knight, Catherine Preston.

**Methodology:** Ruth Knight, Catherine Preston.

**Project administration:** Ruth Knight.

**Resources:** Ruth Knight.

**Software:** Ruth Knight.

**Supervision:** Catherine Preston.

**Validation:** Ruth Knight, Catherine Preston.

**Visualization:** Ruth Knight, Catherine Preston.

**Writing – original draft:** Ruth Knight, Catherine Preston.

**Writing – review & editing:** Ruth Knight, Catherine Preston.

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
