## [Decision Letter · Decision Letter 0]

3 Mar 2023

PONE-D-22-21961Selfies make you look slimmer: the effect of viewing angle on aesthetic and weight judgments of bodiesPLOS ONE

Dear Dr. Knight,

Thank you for submitting your manuscript to PLOS ONE. After careful consideration, we feel that it has merit but does not fully meet PLOS ONE’s publication criteria as it currently stands. Therefore, we invite you to submit a revised version of the manuscript that addresses the points raised during the review process.

We look forward to receiving your revised manuscript.

Kind regards,

Federica Scarpina, Ph.D.

Academic Editor

PLOS ONE

Journal Requirements:

2. Please provide additional details regarding ethical approval and participant consent in the body of your manuscript. In the Methods section, please ensure that you have specified (1) the name of the IRB/ethics committee that approved your study, (2) whether consent was informed and (3) what type of consent you obtained (for instance, written or verbal). If your study included minors, state whether you obtained consent from parents or guardians. If the need for consent was waived by the ethics committee, please include this information.

4. Please ensure that you include a title page within your main document. You should list all authors and all affiliations as per our author instructions and clearly indicate the corresponding author.

Additional Editor Comments (if provided):

I thank you for your patience. As you known, multiple reviewers have been approached, but with no success. I am sorry if this delay causes any issues to your work.

Neverthless, I can send you the comments about your paper from two experts. They both agreed about the need of a profound review of your manuscript, especially in requiring an improvement of the readibility. I strongly suggest you to look at their comments and to work on the paper accordingly.

Best regards,

FS

Reviewers' comments:

Reviewer's Responses to Questions

**Comments to the Author**

1. Is the manuscript technically sound, and do the data support the conclusions?

Reviewer #1: Partly

Reviewer #2: Partly

2. Has the statistical analysis been performed appropriately and rigorously? 

Reviewer #1: Yes

Reviewer #2: No

3. Have the authors made all data underlying the findings in their manuscript fully available?

Reviewer #1: No

Reviewer #2: No

4. Is the manuscript presented in an intelligible fashion and written in standard English?

Reviewer #1: Yes

Reviewer #2: Yes

5. Review Comments to the Author

Reviewer #1: The authors of the present MS investigated the impact of viewing perspective on the perceived attractiveness and body weight based on selfies. The presented data was further analyzed with respect to own body perception and disordered eating behavior. Results provide new insides regarding this relationship and thus contributes valuable information to this research topic. The author’s postulations are supported by convincing data, and I would certainly consider this study suitable as a PONE article. I really appreciate that the authors considered power analyses since I generally find this necessary in experimental studies. Unfortunately, most of the MS I review don’t mind this important step in data analysis. However, I also feel that the manuscript should be improved in several ways for publication as I still see some major methodical and theoretical weaknesses. I provide some hints which potentially help to improve the MS. However, I would also appreciate if the authors also critical respond to my thoughts on their work. Not every point must be addressed (even if I highly recommend to do so).

Main Problems:

1. The title of the MS should be reconsidered, imho. The main reason is that the authors only used female models and the title clearly suggests that results can be generalized. Moreover, the authors do not even discuss this point. My suggestion to solve this problem: Change the title so that the reader is aware of this fact or just add this to your limitations

2. Personally, I found it very interesting and convincing that past research (e.g., Schneider & Carbon 2017 as well as Schneider, Hecht & Carbon, 2012) did not use full-body selfies and that it might be important to combine faces + bodies since this could be more naturalistic (page 10 in your MS). Unfortunately, the authors limited their stimulus material to body only depictions. My suggestion: Make the (potential?) strength of your approach more clear. Otherwise, discuss why you did not follow your own argumentation (weaknesses).

3. The authors did use four viewing angles. I found it hard to understand why they used these perspectives. Is this approach derived from scientific research? Frontal (“allocentric”) is reasonable. However, why is the frontal depiction defined as a selfie? The authors should provide information why only this perspective can be seen as a selfie (in contrast to the “selfie-stick”-condition). There is scientific evidence that other perspectives also can be defined as selfies + have stronger impact on perceived body related factors (see e.g., Schneider & Carbon, 2017).

4. The used material is non-standardized. Past research could show that even small variations of angle might affect the perceived body weight.

5. How did the authors control that every picture was taken from exact the same angle? If the authors belief that this is not necessary, please convince us!

6. Why did only female participants take place in this study (page 20)? Potential cross-gender differences are not discussed

7. I would strongly recommend reconsidering and revising the structure of the MS. I found it puzzling in some way and therefore very hard to follow all procedures, methods, and reported result My suggestion is to split everything off into EXP 1, EXP2 etc. Regarding the results one solution could be to revise the main figures (which were lacking from important information). For a better readability the authors could add some information like effect sizes or/an p-values. This might reduce very detailed data reporting in the plain text.

8. Page 21: From here the authors are consequently talking about their hypothesis. I couldn’t find a list of all hypotheses. It is obvious that some of these hypotheses are derived from the author’s theoretical framework. But this remains unclear.

9. I find it interesting that the authors revealed that the selfie condition yielded lower body weight estimations compared to the “selfie-stick” condition (shown in Fig 3). However, explanation is missing.

Minor problems:

1. Even if I appreciate that the authors did consider power analyses, please stay consequent and report on the verification of ALL ANOVA assumptions to convince us you can use it (e.g., independence of observations, normality of distribution of residuals as well as the homoscedasticity across and within all groups).

2. Data reporting is not consistent across the MS and is not in line with the common APA 6th rules (e.g., SD, M or sometimes p is not italic)

3. Please provide line numbers! As a reviewer it is impossible to be efficient without line numbers.

4. A key reference is missing regarding weight perception and viewing angle: Schneider & Carbon (2012)

References:

Schneider, T. M., & Carbon, C. C. (2017). Taking the perfect selfie: Investigating the impact of perspective on the perception of higher cognitive variables. Frontiers in Psychology, 8(971), 1-16. doi:10.3389/fpsyg.2017.00971

Schneider, T. M., Hecht, H., & Carbon, C. C. (2012). Judging body weight from faces: The height-weight illusion. Perception, 41(1), 121-124. doi:10.1068/p7140

Reviewer #2: PLOS ONE (PONE-D-22-21961)

Selfies make you look slimmer: the effect of viewing angle on aesthetic and weight judgments of bodies.

In this study the authors investigated the role of the perspective of images of the body on attractiveness and weight ratings, specifically considering social media-style images such as selfies. Also, they explored whether these judgments were related to eating disorders thoughts and behaviors. Overall, the results suggest that selfies images were judged to be slimmer while images of body from an egocentric perspective were the least attractive. Also, ratings concerning selfies images were related to eating disorders thoughts and behaviors. The authors briefly discuss the possible implications of these results concerning body satisfaction and eating disorders vulnerability, especially considering the widespread use of social media (i.e., high exposure to selfies-style images of other bodies).

Overall, I found the topic interesting and worth to be investigated considering the widespread use of social media (and thus selfies-style images), especially among the youngest.

However, I have some crucial considerations about the article writing and structure, which in my opinion is quite confusing and not balanced. I believe certain sections are unnecessarily too long and other not sufficiently developed, some pieces of information are redundant and other lacking. I believe these issues significantly affect readability, at the expense of the possibility to get the actual relevance of the results and shading the quality of the research conducted.

I provide my considerations in detail section by section.

Abstract

In my opinion, the abstract might be more concise: e.g., the first sentence is not really needed, and I find the third period (“However, the influence…”) confusing, possibly too long; also, it is not immediately clear what the images perspective are supposed to influence (I imagine the authors refer to attractiveness and weight, as reported afterwards), it may be of help to the reader reporting this more explicitly.

Introduction

1. I found quite difficult to get key information in the Introduction. Personally, I would benefit from reporting more concise and less redundant information, thus, shortening the overall length of this section.

For example, when reporting previous results I would appreciate more concise sentences, which “get to the point” of findings: e.g., “Taking and posting selfies is a popular activity, particularly among social media users, with some participants in one study taking over eight selfies a day (Balakrishnan & Griffiths, 2017)” (page 7) might be something like “Social media users can even take 8 selfies per day, suggesting this is a popular activity ”.

Also, “Donaghue and Smith (2008) explored whether this extends to other physical attributes, such as attractiveness and sexiness, in their study with 60 participants (30 men and 30 women, aged 18 - 88 years) viewing 20 full length photos from an allocentric perspective, including photos of themselves. Participants made self-enhancing judgements of their own attractiveness and sexiness, but self-deprecating judgements of their own body size - they rated themselves as more overweight than ratings made of them by others (Donaghue & Smith, 2008)” (page 11) may be reported in a more incisive form.

Moreover, I believe that the digression and evidence about pictures of faces (page 10), in this current form, seem not relevant since the picture used in the study are of the body only. In fact, I expected evidence about faces vs bodies would be used to support the choice of using images of bodies without faces.

Or, “Firstly, based on previous research that found that selfies of faces were judged as more attractive, egocentric images were judged as more attractive and slimmer only for overweight bodies and that selfies allow for optimal viewing angle which is likely to be slim given the link between attractiveness and slimness in females, it is hypothesised that selfies will be deemed more attractive and slimmer than both allocentric and egocentric images of the same bodies.” (page 13). This is too long to follow, I suggest rewording.

2. “Recent research indicates that viewing more selfies is linked to facial dissatisfaction, a relationship mediated by appearance comparisons (Yang, Fardouly, Wang & Shi, 2020)” (page 12).

Could the authors clarify what does it mean “mediated by appearance comparisons”?

3. Page 12, on the meta-analysis by Ferguson (2013): I apologize, but it is hard to understand the point made about effect size, could the authors clarify this point and clearly link this digression to their own study? Also, beyond the issue concerning effect size, it may be noted that the metanalysis mentioned is quite dated and, conceivably, the studies included are even older; however, the use of social media changed rapidly. I strongly encourage the authors to provide more recent evidence on this issue (e.g., at a first look on pub med the following might be of interest: 10.3390/ijerph20043484 and 10.3390/ijerph16214177). This may give stronger support to the authors’ hypothesis of a correlation between EDs and social-media images ratings (also, please note that eating disorders were initially abbreviated in EDs and then ED was used). Also, I found information in support of this hypothesis a bit scattered across the introduction. I think the purposes of the study may be clearer focusing first on the perspective-related evidence, which it seems to me the main aim of the study and how the authors addressed this research question; then, focusing on the evidence in support of the hypothesis of a possible ED/ratings link, thus introducing the secondary aim of the study and how it was probed. Then, the aims and hypothesis of the study may be summarized all together at the end of the introduction.

4. I suggest reporting the full name of the EDE-Q score (page 13)

5. Page 13 digression on ecological validity: I totally agree with the authors on this point, but I believe it may be more suitable discussing this issue later, among the study possible limitations.

6. Finally, I was surprised by the fact that women with more optimal WHR ratios were judged as less attractive, I wonder whether the authors have a cue about that (just curiosity!)

Methods

Overall, I found the organization of this section quite confusing.

7. Reporting power analysis before stating the procedure and the analyses performed seems awkward since this are key elements for establishing the sample size.

8. Also, In the Participants section I expected to see the overall (i.e., common to the four experiments) inclusion/exclusion criteria and the information on recruitment modalities, which are report later on (e.g., some information is reported in the Procedure). I suggest reporting here the information which are common to the four experiments and then add study-specific information when talking about each of the experiment. Moreover, I think it may be more logical to report the recruitment procedure in the Methods and then results about the sample in the Results section (after the sample size calculation). I understand the authors reported here the power analysis to introduce the sample; however, in my opinion, the structure I suggest may improve readability, please consider this format.

9. Please, clarified whether only females were recruited.

10. Page 16 (Materials): when describing the EDE-Q authors mentioned four subscales (Restraint, Eating Concern, Shape Concern, and Weight Concern). However, when reporting the Results it seem that Shape and Weight Concerns were merged. I could not find any mention/explanations of this. Could the authors clarify this point? Also, could the authors make explicit the reason why the six “additional” items were not taken into consideration in this study?

11. Page 18: the authors reported that “age, gender, and nationality were asked: were male participants excluded? Also, was nationality an exclusion criterion? Finally, were the height/weight measurements asked after the task because the author believe asking for one’s own weight before the task possibly affect ratings?

12. Page 18: Please, provide more details about the procedure. For example, when stating that “the order of these blocks was randomized” do the authors mean the judgment required (attractiveness vs weight) was randomized across participants, thus images were judged concerning attractiveness first and weight after that (and vice versa) whereas “perspectives” was randomized within each block? Were judgments (attractiveness vs weight) repeated four time in each participant for each image? This was true also in experiment four (it is not clear whether the block was repeated four times)? How many trials were included overall? There were constraints or a specific timing relative to the images presentation? Could the authors provide information about the features of the images (e.g., size?). I imagine the size/format of the images depend on the device used: do the authors control for this factor? If not, I suggest discussing the possible implication of this point among the study limitations. For instance, viewing an image on a 15” laptop may be different than viewing them on a smartphone, do the authors believe that this possibly affect ratings? Overall, I think this section may be clearer reporting the information common to the four experiments only once and then focusing on differences (e.g., kind of pictures included).

13. Also, please note that the name of EDE-Q is reported inconsistently across the manuscript (sometimes reporting the version 6.0 and others not).

14. Finally, please clarify whether in experiment four the EDE-Q was not collected, was it because the participants were those (i.e., some of) included in the other experiments?

15. Analysis: I strongly encourage the authors in making separated paragraphs concerning experiments one to three and the fourth since analyses were different and it is not clear what was done in which experiment (e.g., in the first sentence authors mentioned EDE-Q subscales – but this was done only in experiments one to three). Furthermore, I would find easier reporting/discussing the “topics” always in the same order across all the sections. For example, starting always with perspective comparisons (in the introduction, then methods, results and discussion) and then introducing the EDE-Q part. This may also help the reader to differentiate more clearly between primary and secondary aims (if there is a hierarchy) and/or follow the contents while reading. This may be true even tough – temporally speaking - the authors computed before the EDE-Q scores and the average ratings. So that, starting with overall consideration about the analyses (e.g., which software was used etc), then presenting the analyses concerning the experimental protocols, separately for experiment one-three and the fourth and finally everything concerning the EDE-Q scores and correlations. In this regard, using parametric tests for Likert-type data may not be the best option, beyond normality assumptions. So that, could the authors explicit their choice, ideally also reporting reference in support (for clarity’s sake).

16. Concerning correlation, authors should clarify what they mean with “differences” in judgments. I would suggest reporting here (rather than later on) how, and why, the authors computed this additional variable. As I understand this is always a difference in attractiveness and weight between the two perspectives adopted; therefore, the authors may state this computation here in the analyses section (when illustrating experiment one to three) and then in each experiment provide only the relative interpretation of the delta (e.g., positive values means….(i.e., perspective 1- perspective 2) etc. to reduce redundant information.

 

Results

Overall, I suggest reporting more details about statistics may improve the perceived strength of the of the results. Specifically:

17. Page 20: when discussing data normality, please clarify the dependent variables considered (instead referring to “results” in general). Also, I would appreciate reporting Shapiro-Wilk p values (for each variable) or at least an overall p value (e.g., all p values < [the highest one]); this may give raw information about the significance of deviation.

18. Page 20: as mentioned, I suggest moving information about the EDE-Q subscale validity when discussing the results of this scale. It may be useful also to report references for the validity cut-off adopted. Moreover, please see my previous consideration about the computation of the subscales: validity is reported for four subscales but correlational results are reported for three subscales (the SWC acronymous is even used, suggesting that the Shape and Weight concerns subscales were somehow merged). Also, please be consistent in the use (or not) of acronymous.

19. When reporting the experiments 1-4 results, it would be advisable to show means also of non-significant effects. If the authors believe this may reduce readability, they may provide all the means in a table (specifying whether rank means are displayed when non-parametric tests are used). In this regards, standard deviations may be always reported.

20. Results about WHR correlations are confused: it is not clear which variables were correlated, differences in WHR were computed between the two perspective? This should be clearly reported in the analysis. Also, data on WHR (beyond correlational statistic) should be reported. This apply also to EDE-Q scores: in this regard, it may be of interest discussing the average scores relative to the clinical cut-off.

21. I strongly suggest keeping the same order of presentation of results for all experiments; also, it would be helpful referring explicitly to the Table 1 (please define BMI units (i.e., kg/m2) here) when describing the sample of each experiment (As previously mentioned, I believe samples should be described for each experiment independently, before showing the experimental results).

22. Experiment 4: please, report more information about the assumptions check. Importantly, the bayes analyses were not introduced – nor justified - in the analysis section. Also, please give interpretation of bayes factor with proper reference (since it may not be acknowledge by all readers). Finally, since the BF was computed, it may be worthy commenting this briefly in the discussion.

23. As previously mentioned, please provide means and SD for multiple comparisons.

24. Figure 2/3: “Egocentric images were rated as the least attractive, with selfie, selfie-stick, and allocentric images rated as the most attractive” I believe it may be better reporting interpretation of the figure in the text and specify what * indicate since some readers may not be familiar with this kind of chart. Also, I suggest providing figures with higher resolution: they seem not defined and the lower border does not seem cut precisely. Possibly, providing original file source instead of “screen shot” (if this was the case) may improve figures quality.

25. Overall, it may be useful to clearly state the shift between attractiveness and weight ratings results.

Discussion

Overall, I found the discussion quite chaotic. I strongly suggest the authors to apply my previous considerations about organization also to this section. Since there are many experiments it may be of help discussing them once a time and then make overall considerations (a possibility is also to provide single a discussion for each experiment and then a “general discussion”, this may guide the reader along the results). Beyond this, I suggest the following structure: initial hypotheses > results > discussion/interpretation relative to hypotheses and previous literature; specifically, this may be done for experimental data, EDE-Q and WHR correlations following the same order adopted in the previous sections.

26. Page 28: “Results suggested that there are clear differences between selfies and egocentric images, with the latter judged as both slimmer and more attractive.” It seems to me that these results are not coherent with those reported previously, could the authors check this point?

27. Also, I believe that the results might be discussed more deeply, relative to hypothesis (why there are/are not confirmed? What does the literature say?).

For example, “Differences between attractiveness and weight judgments for selfie sand images taken with a selfie-stick were less clear and not replicated across experiments, with experiment three suggesting that selfie-stick images are more attractive than regular selfies and experiment four suggesting selfies may be judged as slimmer than selfie-stick images.” (page 28) how the authors may explain this incongruency?

And (page 29): “Apparent WHR differences across perspectives did correlate with differences in attractiveness for selfies compared to both selfies taken with selfie-sticks and egocentric images, but not relative to allocentric images.” Could the authors elaborate on this? E.g., what this result may suggest?

(page 29) I found confusing referring to something “discussed further below”: grouping “topics” may help the reader instead provide “scatter” discussions.

28. Page 31: as previously pointed out, I found discussing evidence concerning the face, without focusing on that in the study were used body images (i.e., without comparing face-related evidence with the study results), quite misleading. Could the authors enhance the link between face-related evidence and their results?

29. Page 32: “Interestingly, changes in slimness did not always have a corresponding change in attractiveness ratings for selfies” do the authors test this hypothesis, or is there evidence in literature?

30. Page 32: please when discussing correlations, please report whether they are positive or negative. Also, the authors state: “The correlations may mean that those who are more vulnerable to eating disorders might be more vulnerable to the effects of social media, such that those experiencing more thoughts and behaviours around disordered eating are more likely to judge social media style images like selfies as more attractive. I found this observation interesting and with high clinical relevance, however, I suggest rewording this sentence since in this form it seems to evoke a causal link between the two variables, which could not be probed with correlational analyses.

31.

32. Concluding I would appreciate if authors could further elaborate on possible clinical/social implications of their results, to enhance the relevance of their study.

6. PLOS authors have the option to publish the peer review history of their article (what does this mean?). If published, this will include your full peer review and any attached files.

Reviewer #1: No

Reviewer #2: **Yes: **Sofia Tagini

---

## [Author Response · Author response to Decision Letter 0]

12 May 2023

Please find attached Response to Reviewers letter.

---

## [Decision Letter · Decision Letter 1]

19 Jun 2023

PONE-D-22-21961R1Do selfies make women look slimmer? the effect of viewing angle on aesthetic and weight judgments of women’s bodies.PLOS ONE

Dear Dr. Knight,

Thank you for submitting your manuscript to PLOS ONE. After careful consideration, we feel that it has merit but does not fully meet PLOS ONE’s publication criteria as it currently stands. Therefore, we invite you to submit a revised version of the manuscript that addresses the points raised during the review process.

We look forward to receiving your revised manuscript.

Kind regards,

Federica Scarpina, Ph.D.

Academic Editor

PLOS ONE

Journal Requirements:

Reviewers' comments:

Reviewer's Responses to Questions

**Comments to the Author**

1. If the authors have adequately addressed your comments raised in a previous round of review and you feel that this manuscript is now acceptable for publication, you may indicate that here to bypass the “Comments to the Author” section, enter your conflict of interest statement in the “Confidential to Editor” section, and submit your "Accept" recommendation.

Reviewer #1: All comments have been addressed

Reviewer #2: (No Response)

2. Is the manuscript technically sound, and do the data support the conclusions?

Reviewer #1: Yes

Reviewer #2: Yes

3. Has the statistical analysis been performed appropriately and rigorously? 

Reviewer #1: Yes

Reviewer #2: No

4. Have the authors made all data underlying the findings in their manuscript fully available?

Reviewer #1: No

Reviewer #2: Yes

5. Is the manuscript presented in an intelligible fashion and written in standard English?

Reviewer #1: Yes

Reviewer #2: Yes

6. Review Comments to the Author

Reviewer #1: Many thanks to the authors for their detailed and constructive responses to the comments raised in my original review -- I think they've done a great job and I believe that the revised MS is significantly improved as a result. I would now accept this manuscript for publication

Reviewer #2: I appreciate the effort the Authors made in revising the manuscript according to the Reviewers’ suggestions. On my side, I am quite satisfied with how the Authors addressed my previous comments: I believe the manuscript has highly benefitted from revisions. However, I may ask for some further revisions, of moderate entity, which I believe would further strength the paper.

Full comments are reported in the attached file: "Reviewer's Suggestions"

7. PLOS authors have the option to publish the peer review history of their article (what does this mean?). If published, this will include your full peer review and any attached files.

Reviewer #1: No

Reviewer #2: **Yes: **Sofia Tagini

---

## [Author Response · Author response to Decision Letter 1]

24 Aug 2023

I attach a Response to Reviewers letter that addresses each of the reviewers' comments in turn.

---

## [Editor Report · Decision Letter 2]

11 Sep 2023

Do selfies make women look slimmer? the effect of viewing angle on aesthetic and weight judgments of women’s bodies.

PONE-D-22-21961R2

Dear Dr. Knight,

We’re pleased to inform you that your manuscript has been judged scientifically suitable for publication and will be formally accepted for publication once it meets all outstanding technical requirements.

Kind regards,

Federica Scarpina, Ph.D.

Academic Editor

PLOS ONE

Additional Editor Comments (optional):

Dear Authors,

I thank you for your efforts in addressing the Reviewers' comments. Moreover, I sincerely want to thank you for your patience during the process, which requires many months. I am sorry for any inconveniences related to this delay.

Best regards,

FS
---

## [Editor Report · Acceptance letter]

18 Sep 2023

PONE-D-22-21961R2 

Do selfies make women look slimmer? the effect of viewing angle on aesthetic and weight judgments of women’s bodies. 

Dear Dr. Knight:

I'm pleased to inform you that your manuscript has been deemed suitable for publication in PLOS ONE. Congratulations! Your manuscript is now with our production department. 

Kind regards, 

on behalf of

Dr. Federica Scarpina 

Academic Editor

PLOS ONE